# PERCEPTUAL SCALES PREDICTED BY FISHER INFORMATION METRICS

**Jonathan Vacher**[*]
MAP5,
Université Paris Cité, CNRS,
F-75006, Paris, France
jonathan.vacher@u-paris.fr

**Pascal Mamassian**
LSP, Département d'études cognitives,
École normale supérieure, PSL University, CNRS,
75005 Paris, France
pascal.mamassian@ens.fr

## ABSTRACT

Perception is often viewed as a process that transforms physical variables, external to an observer, into internal psychological variables. Such a process can be modeled by a function coined *perceptual scale*. The *perceptual scale* can be deduced from psychophysical measurements that consist in comparing the relative differences between stimuli (*i.e.* difference scaling experiments). However, this approach is often overlooked by the modeling and experimentation communities. Here, we demonstrate the value of measuring the *perceptual scale* of classical (spatial frequency, orientation) and less classical physical variables (interpolation between textures) by embedding it in recent probabilistic modeling of perception. First, we show that the assumption that an observer has an internal representation of univariate parameters such as spatial frequency or orientation while stimuli are high-dimensional does not lead to contradictory predictions when following the theoretical framework. Second, we show that the measured *perceptual scale* corresponds to the transduction function hypothesized in this framework. In particular, we demonstrate that it is related to the Fisher information of the generative model that underlies perception and we test the predictions given by the generative model of different stimuli in a set a of difference scaling experiments. Our main conclusion is that the *perceptual scale* is mostly driven by the stimulus power spectrum. Finally, we propose that this measure of *perceptual scale* is a way to push further the notion of perceptual distances by estimating the perceptual geometry of images *i.e.* the path between images instead of simply the distance between those.

## 1 INTRODUCTION

**Difference Scaling** Difference scaling methods allow us to measure the relative perceptual differences of multiple stimuli in human observers. Such methods have been used as early as in the 1960s to measure the relative differences of perceived color, contrast or loudness (see Maloney & Yang (2003) and references therein). This is only at the beginning of our century that a fitting method, called Maximum Likelihood Difference Scaling (MLDS), was developed (Maloney & Yang, 2003; Knoblauch & Maloney, 2008) to infer the function that maps the physical to the perceptual space. This function is called *perceptual scale*. The critical assumption behind the fitting methods dates back to Thurstone's law of comparative judgment (see case V Thurstone (1927)): the difference between two values along a psychological dimension is corrupted by noise that has a constant variance. The perceptual scale informs us about how a stimulus is perceived when modified along a continuous physical scale (*e.g.* color, contrast, ...). When the slope of the perceptual scale is steep, perception changes rapidly with small physical changes *i.e.* the observer is highly sensitive to physical variations. When the slope is shallow, perception is stable even for large physical variations *i.e.* the observer is weakly sensitive to physical variations. Recently, the MLDS method has been used to measure the perceptual scales of surface texture (Emrith et al., 2010), watercolor effect (Devinck & Knoblauch, 2012), slant-from-texture (Aguilar et al., 2017), lightness (Aguilar & Maertens, 2020) or probabilities (Zhang et al., 2020). However, perceptual scales of more fundamental physical variables such as orientation and spatial frequency have not been measured nor related to existing probabilistic

---

[*]https://jonathanvacher.github.io/

theory of perception. Additionally, while relations between standard Two-alternative Forced Choice (2AFC) measurements and perceptual scales have been studied (Aguilar et al., 2017), it was a theory of perception that was previously described (Wei & Stocker, 2017) that has been used to derive predictions.

**Probabilistic modeling of perception** Thurstone's law of comparative judgment is a first brick in the history of probabilistic modeling of perception (Thurstone, 1927). Indeed, it introduced the incipient concept of random variable (Chebyshev, 1867; Kolmogoroff, 1933) in psychophysics. Then, the development of computer science and information theory had major impact in perception studies, bringing concepts such as redundancy reduction and information maximization (Attneave, 1954; Barlow et al., 1961). More specifically, when applied to texture perception, these concepts led to Julesz' hypothesis that perception of textures is statistical (Victor et al., 2017) *i.e.* textures with similar statistics are indistinguishable. Later on, together with advances in image processing, Julesz' hypothesis led to modern texture synthesis algorithms (Portilla & Simoncelli, 2000; Gatys et al., 2015). In parallel, a core theorem of probabilities, namely Bayes rule, was found to efficiently predicts human perceptual behaviors (Knill & Richards, 1996). Further works have been dedicated to solve the inverse problem of identifying observers' prior that best explain their perception (Stocker & Simoncelli, 2006; Girshick et al., 2011; Vacher et al., 2018; Manning et al., 2023). Inspired by neural population coding models, an optimal observer theory is now described in details by Wei & Stocker (2017). The main consequence of this theory is the existence of a simple relation between perceptual bias and sensitivity. Yet, the theory is limited to a hypothesized scalar perceptual variable while it is established that only part of the neurons of the primary visual cortex is tuned to scalar variables such as spatial and temporal frequencies or orientation (Olshausen & Field, 2005). In higher visual areas, it is more and more difficult to identify scalar variables that uniquely drive single neurons as they respond to increasingly complex patterns (Bashivan et al., 2019). In some previous work, Wainwright (Wainwright, 1999) have used higher dimensional natural image statistics (auto-correlation and power spectrum) to explain various psychophysical observations. Though, it is unclear how this approach relates to the univariate Bayesian framework.

**Fisher information in neural populations** The theory behind the work of Wei & Stocker (2017) is largely inspired by previous work on neural population coding (Brunel & Nadal, 1998). In this and subsequent works (Yarrow et al., 2012; Kanitscheider et al., 2015; Bethge et al., 2002; Wei & Stocker, 2016), it is often ultimately assumed that neurons are Poisson firing neurons parameterized by the tuning curve of a scalar variable. Variants of this optimal coding framework have been explored to explain the response of single neurons Laughlin (1983); von der Twer & MacLeod (2001). As stated previously, it is a quite restrictive framework as all neurons are not tuned to a scalar variable (Olshausen & Field, 2005). By applying this framework to perception, Wei and Stocker remove these unnecessary assumptions. They derive the relation between perceptual bias and sensitivity which at its core comes from the relation between Fisher information and prior under the optimal coding assumption (Brunel & Nadal, 1998)

$$\mathbb{P}_S(s) \propto \sqrt{\mathcal{I}(s)} \tag{1}$$

where $S$ is a stimulus variable. Fisher information is used to quantify the variance of a stimulus estimator from a neural population encoding (Cramer-Rao lower bound). In contrast, priors are introduced in observer models to explain perceptual biases. Therefore, Equation (1) links neural population models and perceptual models. However, less attention is dedicated to the underlying encoding model, where a stimulus variable $S$ is non-linearly related to an internal measurement $M$ through a function $\psi$ plus an additive Gaussian noise $N$ with constant variance,

$$M = \psi(S) + N. \tag{2}$$

Interestingly, such an encoding model is very similar to the assumptions behind the observer model underlying the MLDS method. However, the precise nature of these internal measurements has so far remained abstract.

**Perceptual distance** A perceptual distance is a score of image quality used to quantify and to compare the performances of image restoration or generation methods. Perceptual distances have been introduced to overcome the limitation of the Signal-to-Noise Ratio (also known as SNR). Indeed, images with similar SNR could vary subjectively in quality when presented to human observers (Wang et al., 2004). The Structural SIMilarity index (SSIM) is a score that is popular to provide a better account of perceptual similarity as compared to SNR. Since then, variations of SSIM

have been proposed to more specific purposes such as estimating photo retouching (Kee & Farid, 2011). However, these scores require to compare the image to be rated to a reference image. In recent years, the success of deep generative modeling have led to the emergence of new scores such as the Inception score (Salimans et al., 2016) or the Fréchet Inception Distance (FID) (Heusel et al., 2017). These scores have in common that they compare the generated image distribution to the true empirical image distribution instead of a generated image to a reference image. In addition, they are based on Deep Neural Network (DNN) features. Overall, it has been shown that such DNN features-based scores are better aligned with human perception than SSIM or SNR for example (Zhang et al., 2018). One possible explanation is that DNN are able to capture high-order image statistics and that as hypothesized in vision, our perception is deeply related to image statistics (see Hepburn et al. (2022) and references in previous paragraphs). Yet, these coined perceptual distances are not exempt of limitations as they could be subject to bias when classes specific features are present or not (Kynkäänniemi et al., 2023). Overcoming these biases will likely require to move away from training by measuring higher-order statistics on the image directly without relying on some learned or random filters (Amir & Weiss, 2021). One other limitation of perceptual distances is that they do not provide any information about how well a model has captured the perceptual geometry of images. Providing a full account of perceptual geometry is more demanding, it requires to compare the path when moving from one image to another and not only their distance along this path.

**Contributions** Our work brings several contributions to overcome the limitations introduced above. First, we explain that a convergence theorem of discrete spot noises is a way to resolve the tension between univariate Bayesian theories of perception (Wei & Stocker, 2017) and the high dimensionality of images (Wainwright, 1999). More precisely, the hypothesis that an observer has a univariate representation of the distribution of the parameter of interest (*e.g.* spatial frequency), is compatible with the assumption that an observer is measuring the spectral energy distribution of the image in that both assumptions leads to similar predictions. Second, we show that the function $\psi$, introduced in Equation 2, can be interpreted as the perceptual scale as measured by a difference scaling experiment. Then, we demonstrate again that this function $\psi$ can be predicted from the Fisher information of the stimulus when using the true distribution of the noisy internal stimulus representation knowing the presented stimulus *i.e.* the distribution of the measurements $M$ that we give explicitly. Therefore, we provide a clear link between theory and experiment. Third, we propose to go further in exploring perceptual distances by estimating how well the geometry of natural images captured by models matches the perceptual geometry. For this purpose, we empirically test the prediction given by the Fisher information of Gaussian vectors and processes in a series of experiments (code and data[1] texture interpolation code[2]) involving stochastic stimuli characterized by their power spectrum or their higher-order statistics captured by VGG-19 (Gatys et al., 2015). In particular, we collapse the high dimensionality of these statistics by interpolating between single textures (Vacher et al., 2020) and we measure the corresponding perceptual scale when going from one texture to another. Finally, we propose the Area Matching Score (AMS) score to quantify the mismatch between the predicted and the measured perceptual scales providing a clear method to evaluate the perceptual alignment between generative image models and human vision.

**Notations** Unless stated differently, upper case letters (*e.g.* $X$) are random variables and lower case letters (*e.g.* $x$) are samples or realizations of those random variables. The probability density at $X = x$ is denoted $\mathbb{P}_X(x)$. Similarly, the conditional probability density at $X = x$ knowing $Y = y$ is denoted $\mathbb{P}_{X|Y}(x, y)$. The set $\mathbb{S} = [s_{\text{init}}, s_{\text{final}}]$ is the stimulus segment.

## 2 METHODS

### 2.1 STOCHASTIC VISUAL STIMULATION

We recall some theoretical results about the artificial textures we use in the current work. Firstly, we use textures that are stationary Gaussian Random Fields (GRFs) fully characterized by their scalar mean and their auto-correlation function (or equivalently their power spectrum *i.e.* the auto-correlation Fourier transform). Interestingly, such GRFs can be seen as the limit of high intensity discrete spot noises. This result allows one to relate the densities of local image features such as orientation and scales to the power spectrum of the image (seen as a GRF). In summary, it provides a link between scalar densities and the high-dimensional Gaussian distribution of the image. We will

---

[1]`https://github.com/JonathanVacher/perceptual_metric`
[2]`https://github.com/JonathanVacher/texture-interpolation`

see in later sections that this result leads both approach to similar predictions for perceptual scales. Secondly, we use naturalistic textures that are obtained by imposing high-order and high-dimensional statistics obtained using VGG-19. However, the result mentioned above and detailed below does not hold for naturalistic textures. It is unknown at this stage if similar results could be obtained with some feature under some (non-linear) transformation.

**Asymptotic Discrete Spot Noise** Let $\xi_0 = (1, 0)$. Let $g_\sigma$ be a Gabor function defined for all $\sigma > 0$ and for all $x \in \mathbb{R}^2$ by $g_\sigma(x) = 1/2\pi \cos(x \cdot \xi_0)e^{-\frac{\sigma^2}{2}\|x\|^2}$. In addition, let $\varphi_{z,\theta}$ be a scaled rotation defined for all $(z, \theta) \in \mathbb{R}_+ \times [0, \pi]$ by $\varphi_{z,\theta}(x) = zR_{-\theta}(x)$ where $R_\theta$ is the rotation of angle $\theta$. Now, let $F_{\lambda,\sigma}$ be a discrete spot noise of intensity $\lambda > 0$ defined as the following random field for all $x \in \mathbb{R}^2$, $F_{\lambda,\sigma}(x) = 1/\sqrt{\lambda}\sum_{k\in\mathbb{N}} g_\sigma(\varphi_{Z_k,\Theta_k}(x - X_k))$ where $(X_k, Z_k, \Theta_k)_{k\in\mathbb{N}}$ are iid random variables. Specifically $(X_k)_{k\in\mathbb{N}}$ is a 2-D Poisson process of intensity $\lambda > 0$ and $(Z_k, \Theta_k)_{k\in\mathbb{N}}$ have densities $(\mathbb{P}_Z, \mathbb{P}_\Theta)$.

**Proposition 1** (Convergence and Power Spectrum). *In the limit of infinite intensity ($\lambda \longrightarrow +\infty$) and pure wave ($\sigma \longrightarrow 0$), $F_{\lambda,\sigma}$ converges towards a Gaussian field $F$ with the following power spectrum for all $\xi \in \mathbb{R}^2$,*

$$\hat{\gamma}(\xi) = \frac{1}{\|\xi\|^2}\mathbb{P}_Z(\|\xi\|)\mathbb{P}_\Theta(\angle\xi)$$

*where $\xi = (\|\xi\|\cos(\angle\xi), \|\xi\|\sin(\angle\xi))$.*

*Proof.* This is a special case of Proposition 2 in Vacher et al. (2018). The general result is Theorem 3.1 in Galerne (2010). $\square$

In practice, the distribution $\mathbb{P}_Z$ and $\mathbb{P}_\Theta$ are parametrized by $(Z_0, \Sigma_Z)$ and $(\Theta_0, \Sigma_\Theta)$ respectively.

By providing a relation between local feature statistics (orientation and scale) and the image power spectrum, Proposition 1 will allow us to justify the common assumption made when modeling psychophysical data that is, the feature of interest is directly used by the observer instead of the image (Knill & Richards, 1996; Stocker & Simoncelli, 2006; Girshick et al., 2011). See Section 2.3.

**Interpolation of Naturalistic Textures** Even though for experimental purposes the GRFs described above can be parameterized by just a few scalar variables (Vacher et al., 2018), naturalistic textures depend on the statistics of high-dimensional features extracted at different layers of VGG-19 (Gatys et al., 2015; Vacher et al., 2020). Previous algorithms widely used in vision studies (Portilla & Simoncelli, 2000; Vacher & Briand, 2021) were using fewer parameters, but the number was still too large to derive clear and interpretable results (Okazawa et al., 2015). One way to efficiently collapse the dimension parameterizing those textures is to use interpolation (Vacher et al., 2020). As a consequence the texture features of an interpolation of textures extracted at layers $k$ are interpreted as realizations of a random variable $A_k(s)$ with mean $\mu_W(s)$ and covariance $\Sigma_W(s)$ (see Appendix C). Assuming Gaussiannity, it will become possible to derive predictions for the perceptual scale measured along the interpolation path.

## 2.2 Thurstone Scale, Fisher Information and MLDS

First, we define more precisely the encoding model given in Equation 2 as follows

$$M = R + N \quad \text{where} \quad R = \psi(S) \quad \text{with} \quad \psi : \mathbb{S} \to \mathbb{S}. \tag{3}$$

We use the above description to highlight the fact that $M$, $R$ and $N$ are random variables that are internal to the observer while $S$ is external to them, it is an environment variable, an external stimulus. In practice, the noise $N$ is often assumed to be Gaussian with variance $\sigma^2$. It corrupts the internal representation of the stimulus $R$ to give what we call the internal measurement $M$. Then, we define Fisher information for two abstracts unidimensional random variables.

**Definition 1** (One-dimensional Fisher information). *Let $X$ and $Y$ be two random variables defined respectively on two abstract spaces $\mathbb{X}$ and $\mathbb{Y}$ and let $\mathbb{P}_{X|Y}$ be the conditional density of $X$ knowing $Y$. The Fisher information carried by $X$ about $Y$ is a function $\mathcal{I} : \mathbb{Y} \longrightarrow \mathbb{R}$ defined for all $y \in \mathbb{Y}$ by*

$$\mathcal{I}_Y(y) = \mathbb{E}_{X|Y}\left(\left(\frac{\partial \log(\mathbb{P}_{X|Y})}{\partial y}(X, y)\right)^2\right). \tag{4}$$

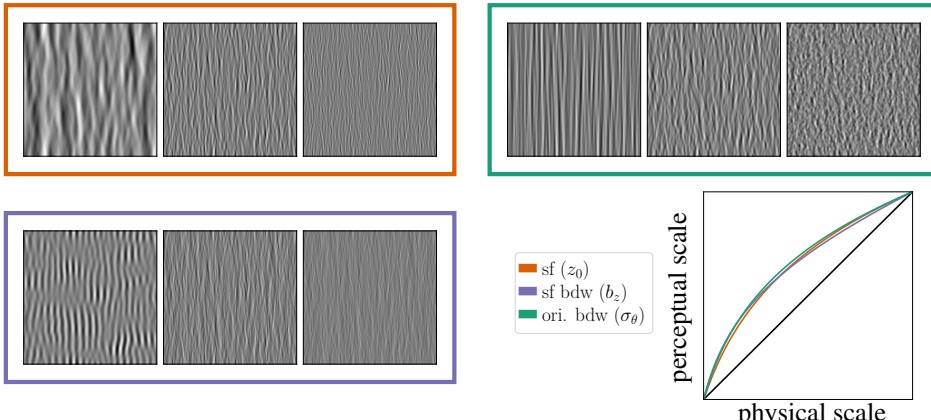

Figure 1: Texture samples and predicted perceptual scales for the spatial frequency mode ($z_0$), the spatial frequency bandwidth ($b_z$) and the orientation bandwidth ($\sigma_\theta$). Bottom-right : prediction obtained by combining Appendix A and Equation (5)

In statistics, Fisher information is used as a upper bound of the precision of an estimation (see Cramér-Rao bound). This is also how we interpret it for an observer, namely the maximal precision of their estimation of a stimulus $S$.

The precise definition given above is helpful to realize that the Fisher information carried by $M$ about $S$ ($\mathcal{I}_S$) is different from the one carried by $M$ about $R$ ($\mathcal{I}_R$). We can go one step further though, and establish the following relation between those two

$$\mathcal{I}_S(s) = \psi'(s)^2 \mathcal{I}_R(\psi(s)). \tag{5}$$

A reformulation of Thurstone law of comparative judgment (Thurstone, 1927) is to assume that the Fisher information of an observer's internal representation $\mathcal{I}_R$ is constant. It is not so obvious to understand why this assumption is relevant. The idea is that an observer has only access to her internal states, she never observes any realization of an external stimulus $S$. Every external variable is transformed to an internal one through the psychological function $\psi$. Therefore, without any knowledge about the external world, a fair assumption is to allocate equal resources to every possible internal state in order to be equally precise in our estimates of different states (without knowing to what they correspond to in the external world). This assumption is also equivalent to assuming that internal observer's noise (a common notion used in psychophysics) is constant. If the internal Fisher information is constant we can now express the psychological function simply in terms of external Fisher information. This is summarized in the following proposition.

**Proposition 2.** *Assume Equation* (3)*, the internal Fisher information $\mathcal{I}_R$ is constant if and only if for all $s \in \mathbb{S}$ the psychological function $\psi$ verifies*

$$\psi(s) \propto \int_{s_{init}}^{s} \sqrt{\mathcal{I}_S(t)} \mathrm{d}t. \tag{6}$$

*Proof.* See Appendix D. □

**Relation to the MLDS observer model.** In the MLDS framework, an observer has to judge which pair of stimuli is more similar to another. Assuming three stimuli ($s_i, s_j, s_k$), those are transformed through the psychological scale $\psi$ and the observer responds by comparing the difference of differences between the pairs $d_{i,j,k} = |\psi(s_i) - \psi(s_j)| - |\psi(s_j) - \psi(s_k)|$. This difference is assumed to be corrupted by an internal noise $N_{\mathrm{mlds}}$ of constant variance $\sigma^2_{\mathrm{mlds}}$ *i.e.* $\Delta_{i,j,k} = d_{i,j,k} + N_{\mathrm{mlds}}$. Those assumptions are sufficient to recover an estimate of $\psi$ (Knoblauch & Maloney, 2008). In addition, it is often assumed that there is no specific internal ordering of the variables so that the difference $d_{i,j,k}$ can be written without absolute value. In that case, assuming the encoding model (3) is enough to recover the MLDS observer model, we have the following relation between the noise variances $\sigma^2_{\mathrm{mlds}} = 4\sigma^2$.

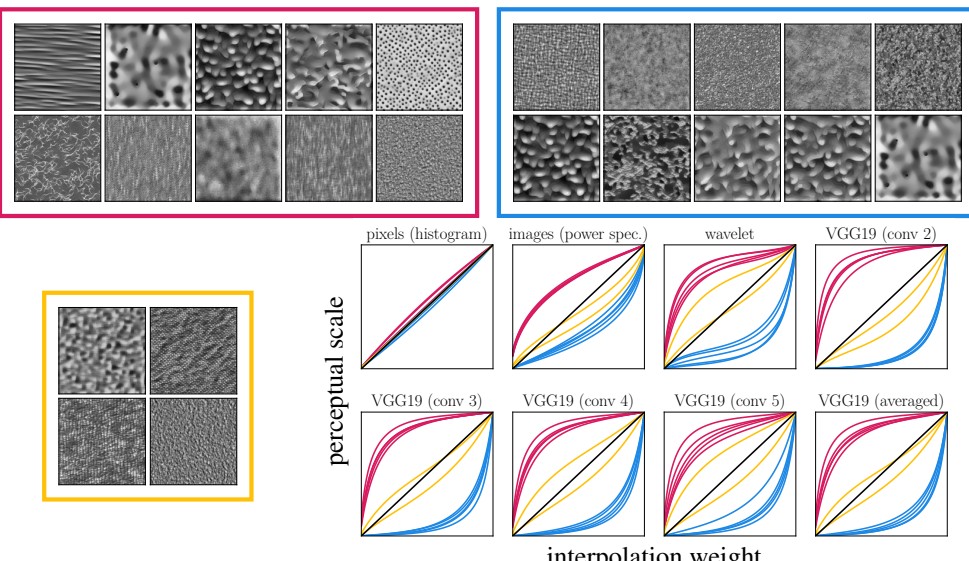

Figure 2: Texture samples and predicted perceptual scales for various interpolation between arbitrary textures. Red corresponds to the early sensitivity group (*i.e.* shallow-to-steep slope). Blue corresponds to the late sensitivity group (*i.e.* steep-to-shallow slope). Yellow corresponds to conflicting prediction across VGG-19 layers. Bottom-right : prediction obtained by combining Proposition 4 and Equation (5). For pixels, images and wavelet, we also assume Gaussiannity as in Equation (8) (pixel and wavelet) and as in Equation (7) (images). See details in Appendix F.

## 2.3   FISHER INFORMATION CARRIED BY THE IMAGE *vs* BY THE LOCAL FEATURES

In the previous section, we have introduce an observer model based on an abstract random internal measurement $M$. It is often unclear what those measurements are. Ideally inspired by neurophysiology, the measurements are responses of neurons to the image often modeled by Linear/Non-linear operations, even though those modeling stages are often dropped in perceptual studies. In the case of GRFs parameterized by spatial frequency (or scale) and orientation distributions, it is often accepted to assume that the measurements are samples of an appropriate distribution *e.g.* a Log-Normal distribution for the spatial frequency or a Von-Mises distribution for the orientation. Using the notation of the previous section, these cases correspond to measurement $M = Z$ with stimulus $S = Z_0$ and measurement $M = \Theta$ with stimuli $S = \Theta_0$. We will see that in both cases this is equivalent to consider that measurements are the image itself $M = F$ with $S = Z_0$ or $S = \Theta_0$. This is because Fisher information is given in closed-form and that perceptual scale can be predicted using Proposition 2. Similar results hold for $S = B_Z$ and $S = \Sigma_\theta$ (note that $(Z_0, B_Z)$ and $(\Theta_0, \Sigma_\Theta)$ are parameters of $\mathbb{P}_Z$ and $\mathbb{P}_\Theta$ introduced in Section 2.1). The predictions are given in Figure 1.

**Fisher Information of Log-Normal and Von-Mises Distributions**   We give the precise parametrization and the corresponding Fisher information of the Log-Normal and Von-Mises distributions in Appendix A.

**Fisher Information of Parametric GRFs**   Now, we consider a GRF texture $F$ with mean $\mu \in \mathbb{R}$ and autocorrelation function $\gamma$ (or equivalently power spectrum $\hat{\gamma}$) parameterized by $s \in \mathbb{S}$. Mathematically, the texture can be expressed for all $x \in \mathbb{R}^2$ and $s \in \mathbb{S}$, as

$$F(x, s) = \mu + \int_{\mathbb{R}^2} k(x - y, s)\mathrm{d}W(y) \tag{7}$$

where $k(\cdot, s) = \mathcal{F}^{-1}(\sqrt{\hat{\gamma}(\cdot, s)})$ and $W$ is a classical Wiener process.

**Proposition 3.** *The Fisher Information carried by $F$ about $S$ is*

$$\mathcal{I}(s) = \frac{1}{2}\int_{\mathbb{R}^2} \frac{1}{|\hat{\gamma}(\xi, s)|^2}\left|\frac{\partial\hat{\gamma}(\xi, s)}{\partial s}\right|^2 \mathrm{d}\xi = \frac{1}{2}\int_{\mathbb{R}^2}\left|\frac{\partial\log(\hat{\gamma}(\xi, s))}{\partial s}\right|^2 \mathrm{d}\xi.$$

*Proof.* This is a specific case of Whittle formula (Whittle, 1953, Theorem 9). □

We combine Proposition 3 with Proposition 1 using the Log-normal and the Von-Mises distributions to express the power spectrum $\hat{\gamma}$ parameterized by $S = Z_0$, $S = B_Z$, $S = \Theta_0$ or $S = \Sigma_\theta$. Therefore, the Fisher information carried by measurements $M = F$ comes down to the Fisher information carried by measurements $M = Z$ (spatial frequency) or $M = \Theta$ (orientation) as described above up to a multiplicative constant of $1/2$. As a consequence both approaches lead to similar predictions about the perceptual scales measured for these parameters.

**Fisher Information of Parametric Gaussian Vectors** In the case of interpolation between naturalistic textures, we do not have a direct generative model of the texture conditionally on the interpolation parameter $s$. Instead, the texture is generated using a gradient descent to impose the statistics of VGG-19 features at multiples layers for which we have a generative model. Therefore, at layer $k$ and for $s \in \mathbb{S}$ the activation $A_k(s)$ of texture $F_k(s)$ is

$$A_k(s) = \mu_k(s) + \Sigma_k(s)N \quad \text{with} \quad \mu_k \in \mathcal{C}^1\left(\mathbb{S}, \mathbb{R}^{d_k}\right) \quad \text{and} \quad \Sigma_k \in \mathcal{C}^1\left(\mathbb{S}, \mathbb{R}^{d_k \times d_k}\right) \tag{8}$$

where $N \sim \mathcal{N}(0, I_{d_k})$ is a standard normal random vector and $d_k$ is the feature dimension of layer $k$.

**Proposition 4.** *The Fisher Information carried by $A_k$ about $S$ is*

$$\mathcal{I}(s) = \mu_k'(s)\Sigma_k(s)^{-1}\mu_k'(s) + \frac{1}{2}\mathrm{Tr}\left(\Sigma_k(s)^{-1}\Sigma_k'(s)\Sigma_k(s)^{-1}\Sigma_k'(s)\right).$$

*Proof.* See Appendix B. □

As stated earlier in the manuscript, no link can be made with interpretable feature distributions as it is the case for GRFs. Though, precise expressions for $\mu_k$ and $\Sigma_k$ are available in close forms in the Gaussian case as assumed here (see Appendix C). In practice, the feature activations are not Gaussian (Vacher et al., 2020).

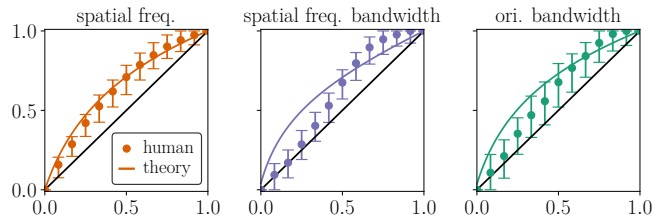

Figure 3: Measured and predicted perceptual scales for the spatial frequency mode (left), the spatial frequency bandwidth (middle) and the orientation bandwidth (right).

### 2.4 PREDICTIONS AND EXPERIMENTAL METHODS

The calculated Fisher informations together with Proposition 2 allows us to predict the perceptual scales corresponding to the parameters described in the previous section. These predictions hold under the assumed generative models for measurements.

**Predictions** In the case of GRFs textures, we recall that assuming that spatial frequency or orientation are directly measured or that the image as a whole is measured makes no differences in the prediction (see bottom-right of Figure 1). In the case of naturalistic textures, the measurements are assumed to be the feature activations of the texture in VGG-19 at layer 2 to 5 (see bottom-right of Figure 2). For the naturalistic, we alternatively propose that measurements are the single pixel gray levels, the image itself (*i.e.* the power spectrum as for GRFs) or the wavelet activations.

**Experimental Methods** The experiment consists of trials where participants have to make a similarity judgment. Participants are presented with 3 stimuli with parameters $s_1 < s_2 < s_3$ and are required to choose which of the two pairs with parameters $(s_1, s_2)$ and $(s_2, s_3)$ is the most similar. We used four sets of textures (see Figure 1 and 2): (i) the first set consists of parameterized artificial textures where we measured the perceptual scales of spatial frequency, spatial frequency bandwidth and orientation bandwidth (see Appendix E for details); (ii) the other three sets consist of interpolations between arbitrary textures. A set of textures for which the perceptual scale corresponds to an early sensitivity (*i.e.* steep-to-shallow slope, see top-left of Figure 2). Another one for which the perceptual scale corresponds to late sensitivity (*i.e.* shallow-to-steep slope, see top-right of Figure 2). And a last set for which the predictions are inconsistent from one layer of VGG-19 to another (bottom-left of Figure 2). All stimuli had an average luminance of 128 (range $[0, 255]$) and an RMS contrast of 39.7. For each texture pair, we use 13 equally spaced ($\delta_s = 0.08\underline{3}$) interpolation weights. To ensure that stimulus comparisons are around the discrimination threshold we only use triplets such that $|s_{1,3} - s_2| \leq 3\delta_s$. For each texture pair, a group of 5 naive participants performed the experiment. Participants were recruited through the platform prolific (https://www.prolific.com), performed the

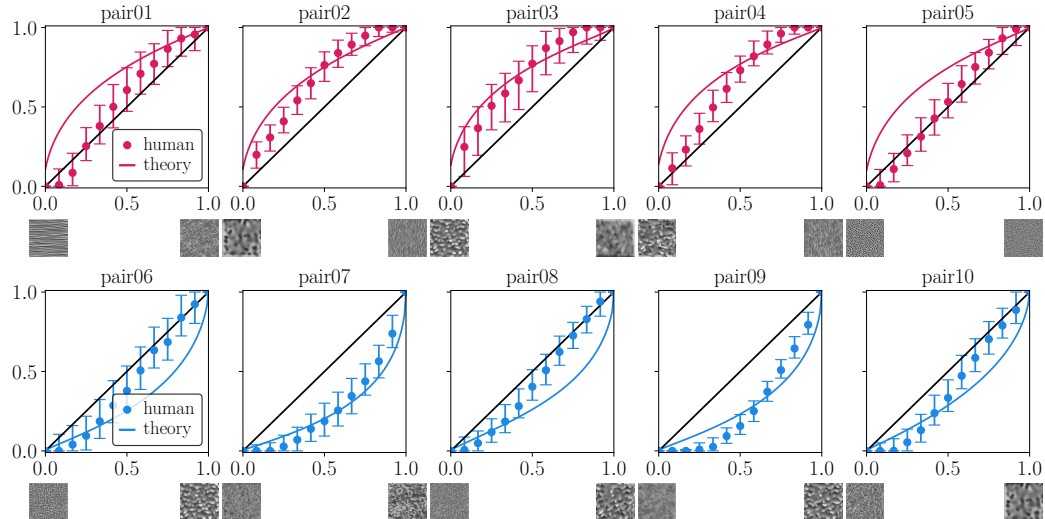

Figure 4: Measured and predicted (power spectrum) perceptual scales for the early (top row) and late (bottom row) sensitivity pairs. Error bars represent $99.5\%$ bootstrapped confidence intervals.

experiments online, and were paid 9£/hr. Monitor gamma was measured using a psychometric estimation and corrected to 1. The MLDS model is described at the end of Section 2.2. The protocol was approved by the CER U–Paris (IRB 00012020–54).

# 3 RESULTS

## 3.1 ORIENTATION AND SPATIAL FREQUENCY

The perception of spatial frequencies is well-known in vision studies, its perceptual scale is expected to be logarithmic. Such a scale is also predicted by Fisher information as integrating the square-root (Proposition 2) of a squared inverse (Proposition 5) leads to a logarithm. The measured perceptual scale of the spatial frequency mode matches correctly this prediction (left of Figure 3). The spatial frequency and orientation bandwidths are less studied, the predictions are qualitatively the same as for the spatial frequency mode (Proposition 2 and Appendix Proposition 6). The Fisher information of the orientation bandwidth

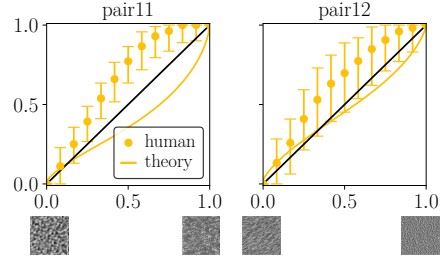

Figure 5: Measured and predicted (auto-cor) perceptual scales for the conflicting prediction pairs. Error bars represent $99.5\%$ bootstrapped confidence intervals.

is more complex but leads to a similar curve. The measured perceptual scale is more variable for the orientation bandwidth (larger error bars) but is still in line with the prediction (predicted offset from linear behavior is exaggerated, see right of Figure 3). In contrast, the measured perceptual scales of spatial frequency bandwidth is approximately linear for low values while its gets supra-linear at intermediate values and even saturate for the highest values (center of Figure 3).

## 3.2 INTERPOLATION BETWEEN NATURALISTIC TEXTURES

We present the perceptual scales measured for the different groups of natural textures in Figure 4 and 5. On these figures, the prediction given by the auto-correlation (*i.e.* when considering the textures as GRFs) is shown. Predictions assuming alternative measurements (pixel, wavelet, and VGG-19) are quantitatively compared in Figure 6 using the following Area Matching Score : AMS = $\int_0^1 \text{sign}(f_m(x)-x)(f_{th}(x)-x)/|f_m(x)-x|\mathrm{d}x$ where $f_m$ and $f_{th}$ are respectively the measured and predicted scales. Intuition about score values is given in Figure 6a.

**Early and Late Sensitivity** For the set of early sensitivity texture pairs, measured perceptual scales are inline with the predictions (Figure 4) in the sense that a linear (`pair01`, `pair04-05`) or a supra-linear (`pair02-03`) perceptual scale is measured in all texture pairs. The same hold for the set of late sensitivity texture pairs (Figure 4), but this time in the sense that a linear (`pair06` and `pair08`) or a sub-linear (`pair07` and `pair09-10`) perceptual scale is measured in all texture

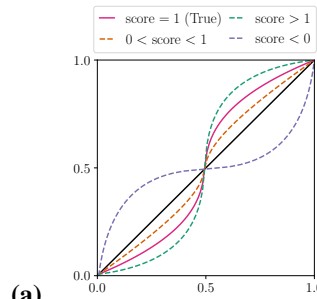
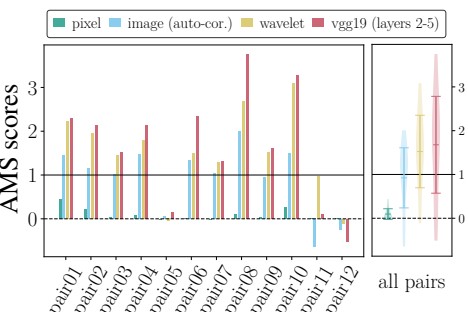

**(a)**    **(b)**

Figure 6: Prediction scores. **(a)** Intuition about score values when comparing two perceptual functions. **(b)** Left : scores for each pair and for different predictions. Right : averaged scores for different predictions with $99.5\%$ bootstrapped confidence intervals.

pairs. Such a result is also valid for the predictions based on alternative measurement assumptions as shown in Figure 6 by the fact that all scores are positive for `pair01-10`.

**Conflicting predictions** For the set of textures with conflicting predictions (Figure 5), for both texture pairs, we observe that the GRF measurement assumption predicts a late sensitivity while the measured perceptual scale corresponds to an early sensitivity with a late saturation. Other measurement assumptions are not providing better prediction as their score is either close to $0$ or negative. However, there is one exception for `pair11` under the wavelet measurement assumption which has a ideal score, close to $1$ (up to score limitation, see Section 4).

**Measurement Assumption Scores** As previously stated, all assumptions predict correctly whether the scale corresponds to late or early sensitivity (positive scores) except for the conflicting prediction texture pairs. Note that `pair05` has also a score close to $0$ under all assumptions (though this might be due to score limitation, see Section 4). On average, the GRF assumption is the best with an average score ($\pm 99.5\%$ CI) close to $1$ ($0.92 \pm 0.69$). The single pixel distribution assumption only predicts a linear behavior and has therefore a score close to $0$. In contrast, the wavelet and VGG-19 assumptions often overestimate the early or late sensitivity (average scores above $1$). We conducted additional experiments in which we fixed the power spectrum of all textures along a path between a pair to be the average of the pair's. In this case the power spectrum cannot explain the measured perceptual scale the operation is indeed deteriorating discriminability (see Appendix H).

## 4    DISCUSSION AND CONCLUSION

In the case of GRFs, we have shown that the univariate assumption behind the Bayesian theories of perception and the absence of this assumption (*i.e.* the observer is using all the information in the image) lead to the same prediction for the perceptual scales of spatial frequencies, orientations and their bandwidths. Such a result is due to the fact that these local feature distributions directly appear in the power spectrum (the Fourier transform of the auto-correlation) of GRFs (Proposition 1) and to our second result that is the perceptual scale is related to the Fisher information of the feature distribution (Proposition 2). In the case of naturalistic textures, it is unknown if such a result relating a (non-linear) transform and some feature distribution holds. Therefore, it is necessary to make new hypotheses about the measurements in order to predict the perceptual scale of an observer. We tested this issue in a series of difference scaling experiments involving GRF and naturalistic textures. Our main result is that the perceptual scale is mainly driven by the auto-correlation (or the power spectrum). However, it does not perfectly explain the measured perceptual scales, and in particular, the perceptual scale of `pair11` appears to be driven by the wavelet representation. A highly interesting future directions is to compare the perceptual scale to a neurometric scale, an equivalent scale but deduced from neurophysiological recordings as the equivalent exists for the psychometric function (Newsome et al., 1989; Berens et al., 2011). Other limitations lie in the MLDS method. Usually, running a difference scaling experiment requires to know ahead of time an approximation of the observer's sensitivity to the parameter that one would like to test. Here, we have not estimated the sensitivity of each participant and, therefore, have not adapted the stimuli accordingly. Yet, it seems that we were near the participants sensitivity (see Appendix G). In addition, the MLDS method is limited to the study of a single stimulus dimension while extensions can still be developed Knoblauch et al. (2012) and compared to higher-dimensional theories Malo & Gutiérrez (2006); Laparra & Malo (2015). All these questions demonstrate the ambition of our approach and the work that remains to be done to understand, beyond perceptual distance, perceptual metrics.

## REPRODUCIBILITY STATEMENT

The reproducibility of our work will be ensured by the links provided to the data and code. Theoretical results are supported by proofs or references to proof.

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

## A  FISHER INFORMATION OF LOG-NORMAL AND VON-MISES RANDOM VARIABLES

A random variable $Z$ follows a log-normal distribution ($Z \sim \mathcal{LN}(z_0, b_Z)$) if it has the following density defined for all $z > 0$ by

$$\mathbb{P}_Z(z; z_0, b_Z) = \frac{2}{z b_Z \sqrt{\ln(2)\pi}} \exp\left(-\frac{4\left(\ln\left(\frac{z}{z_0}\right) - \frac{\ln(2)}{8}b_Z^2\right)^2}{\ln(2)b_Z^2}\right)$$

where $z_0$ is the mode and $b_Z$ is the octave bandwidth of the density. A random variable $\Theta$ follows a Von-Mises distribution ($\Theta \sim \mathcal{VM}(\theta_0, \sigma_\Theta)$) if it has the following density defined for all $\theta \in [0, \pi]$ by

$$\mathbb{P}_\Theta(\theta; \theta_0, \sigma_\Theta) = \frac{1}{\pi \mathcal{B}_0(1/4\sigma_\Theta^2)} \exp\left(\frac{\cos\left(2(\theta - \theta_0)\right)}{4\sigma_\Theta^2}\right).$$

**Proposition 5.** *Let $Z \sim \mathcal{LN}(z_0, b_Z)$. The Fisher Informations carried by $Z$ about respectively $Z_0$ and $B_Z$ are*

$$\mathcal{I}(z_0) = \frac{8}{z_0^2 b_Z^2 \ln(2)} \quad and \quad \mathcal{I}(b_Z) = \frac{\ln(2)}{2}\left(1 + \frac{4}{b_z^2 \ln(2)}\right)$$

**Proposition 6.** *Let $\Theta \sim \mathcal{VM}(\theta_0, \sigma_\Theta)$. The Fisher Informations carried by $\Theta$ about respectively $\Theta_0$ and $\Sigma_\Theta$ are*

$$\mathcal{I}(\theta_0) = \frac{1}{\sigma_\Theta^2}\frac{\mathcal{B}_1(1/4\sigma_\Theta^2)}{\mathcal{B}_0(1/4\sigma_\Theta^2)} \quad and \quad \mathcal{I}(\sigma_\Theta) = \frac{1}{4\sigma_\Theta^6}\left(1 - \frac{\mathcal{B}_1(1/4\sigma_\Theta^2)}{\mathcal{B}_0(1/4\sigma_\Theta^2)}\left(4\sigma_\Theta^2 + \frac{\mathcal{B}_1(1/4\sigma_\Theta^2)}{\mathcal{B}_0(1/4\sigma_\Theta^2)}\right)\right)$$

# B  FISHER INFORMATION OF PARAMETRIC GAUSSIAN VECTOR

*Proof of Proposition 4.* We start by reminding the following calculus results (Petersen et al., 2008):

(i) $\nabla \log(|A|) = \left(A^{-1}\right)^T$,

(ii) $\mathrm{d}A^{-1} = -A^{-1}\mathrm{d}AA^{-1}$.

The log of $\mathbb{P}_{X|S}$ is

$$\log(\mathbb{P}_{X|S}(x|s)) = -\frac{1}{2}\log(2\pi) - \frac{1}{2}\log(|\Sigma(s)|) - \frac{1}{2}(x - \mu(s))^T\Sigma(s)^{-1}(x - \mu(s)).$$

Then,

$$\frac{\partial \log(\mathbb{P}_{X|S})}{\partial s}(x, s) = -\frac{1}{2}\mathrm{Tr}(\Sigma(s)^{-1}\Sigma'(s)) - \frac{1}{2}(x - \mu(s))^T\left(\Sigma(s)^{-1}\right)'(x - \mu(s))$$
$$+ (x - \mu(s))^T\Sigma(s)^{-1}\mu'(s).$$

Then,

$$\left(\frac{\partial \log(\mathbb{P}_{X|S})}{\partial s}(x, s)\right)^2 = C_1 + C_2 + C_3 + C_4 + C_5 + C_6$$

where

$$C_1 = \frac{1}{4}\mathrm{Tr}(\Sigma(s)^{-1}\Sigma'(s))^2,$$
$$C_2 = \frac{1}{4}\left((x - \mu(s))^T\left(\Sigma(s)^{-1}\right)'(x - \mu(s))\right)^2,$$
$$C_3 = \left((x - \mu(s))^T\Sigma(s)^{-1}\mu'(s)\right)^2,$$
$$C_4 = \frac{1}{2}\mathrm{Tr}\left(\Sigma(s)^{-1}\Sigma'(s)\right)(x - \mu(s))^T\left(\Sigma(s)^{-1}\right)'(x - \mu(s)),$$
$$C_5 = -\mathrm{Tr}\left(\Sigma(s)^{-1}\Sigma'(s)\right)(x - \mu(s))^T\Sigma(s)^{-1}\mu'(s),$$
$$C_6 = -(x - \mu(s))^T\Sigma(s)^{-1}\mu'(s)(x - \mu(s))^T\left(\Sigma(s)^{-1}\right)'(x - \mu(s)).$$

First, we observe that $\mathbb{E}(C_1) = C_1$ and that $\mathbb{E}(C_5) = \mathbb{E}(C_6) = 0$ (odd central moments). Then, following (ii), we have

$$\left(\Sigma(s)^{-1}\right)' = -\Sigma(s)^{-1}\Sigma'(s)\Sigma(s)^{-1}.$$

Hence,

$$\mathbb{E}\left((x - \mu(s))^T\left(\Sigma(s)^{-1}\right)'(x - \mu(s))\right) = -\mathbb{E}\left((x - \mu(s))^T\Sigma(s)^{-1}\Sigma'(s)\Sigma(s)^{-1}(x - \mu(s))\right)$$
$$= -\mathbb{E}\left(\left(\sqrt{\Sigma'(s)}\Sigma(s)^{-1}(x - \mu(s))\right)^T\left(\sqrt{\Sigma'(s)}\Sigma(s)^{-1}(x - \mu(s))\right)\right)$$
$$= -\mathrm{Tr}(\Sigma(s)^{-1}\Sigma'(s)).$$

[Note : think about the covariance of $\sqrt{\Sigma'(s)}\Sigma(s)^{-1}(x - \mu(s))$ or see Magnus (1978).]
Therefore, the expectation of $C_4$ is

$$\mathbb{E}(C_4) = -\frac{1}{2}\operatorname{Tr}(\Sigma(s)^{-1}\Sigma'(s))^2.$$

The expectation of $C_2$ is the second order moment of a quadratic form (see Magnus (1978)). Thus,

$$\mathbb{E}(C_2) = \frac{1}{4}\operatorname{Tr}(\Sigma(s)^{-1}\Sigma'(s))^2 + \frac{1}{2}\operatorname{Tr}(\Sigma(s)^{-1}\Sigma'(s)\Sigma(s)^{-1}\Sigma'(s)).$$

We can now deal with the expectation of the last term $C_3$,

$$\mathbb{E}(C_3) = \mathbb{E}\left(\left((x - \mu(s))^T\Sigma(s)^{-1}\mu'(s)\right)^2\right) = \mathbb{E}\left((x - \mu(s))^T\Sigma(s)^{-1}\mu'(s)(x - \mu(s))^T\Sigma(s)^{-1}\mu'(s)\right)$$

$$= \mathbb{E}\left(\mu'(s)^T\Sigma(s)^{-1}(x - \mu(s))(x - \mu(s))^T\Sigma(s)^{-1}\mu'(s)\right)$$

$$= \mu'(s)^T\Sigma(s)^{-1}\mathbb{E}\left((x - \mu(s))(x - \mu(s))^T\right)\Sigma(s)^{-1}\mu'(s) = \mu'(s)^T\Sigma(s)^{-1}\mu'(s).$$

Summing all the expectations leads to the result.

$\square$

## C   MEAN AND COVARIANCE OF A WASSERSTEIN BARYCENTER

**Proposition 7** (Wasserstein Barycenter of Gaussian Distributions). *Let $X_0 \sim \mathcal{N}(\mu_0, \Sigma_0)$ and $X_1 \sim \mathcal{N}(\mu_1, \Sigma_1)$ be two Gaussian random variables. The Wasserstein interpolation between $\mathbb{P}_{X_0}$ and $\mathbb{P}_{X_1}$ is defined as*

$$\mathbb{P}_s = \operatorname*{argmin}_{\mathbb{P} \in \mathcal{P}(\mathbb{P}_{X_0}, \mathbb{P}_{X_1})} (1 - s)W_2(\mathbb{P}_{X_0}, \mathbb{P})^2 + sW_2(\mathbb{P}_{X_1}, \mathbb{P})^2$$

*where $\mathcal{P}(\mathbb{P}_{X_0}, \mathbb{P}_{X_1})$ is the set of probability densities with marginals $\mathbb{P}_{X_0}$ and $\mathbb{P}_{X_1}$ and where*

$$W_2(\mathbb{P}_X, \mathbb{P}_Y)^2 = \inf_{X \sim \mathbb{P}_X, Y \sim \mathbb{P}_Y} \mathbb{E}\left(\|X - Y\|^2\right)$$

*is the Wasserstein distance between $\mathbb{P}_X$ and $\mathbb{P}_Y$. The probability distribution $\mathbb{P}_s$ is Gaussian with mean $\mu_W(s) = (1 - s)\mu_{X_0} + s\mu_{X_1}$ and covariance*

$$\Sigma_W(s) = \Sigma_{X_0}^{-1/2}\left((1 - s)\Sigma_{X_0} + s\left(\Sigma_{X_0}^{1/2}\Sigma_{X_1}\Sigma_{X_0}^{1/2}\right)^{1/2}\right)^2 \Sigma_{X_0}^{-1/2}.$$

*Proof.* See (Chen et al., 2018). $\square$

**Corollary 1.** *Let $X_s \sim \mathcal{N}(\mu_W(s), \Sigma_W(s))$ where $((\mu_W(s), \Sigma_W(s)))$ are defined in Proposition 7, then the Fisher information is given by*

$$\mathcal{I}(s) = (\mu_{X_1} - \mu_{X_0})^T\Sigma_{WB}(s)^{-1}(\mu_{X_1} - \mu_{X_0})$$

$$+ \frac{1}{2}\operatorname{Tr}\left(\Sigma_{WB}(s)^{-1}\Sigma'_{WB}(s)\Sigma_{WB}(s)^{-1}\Sigma'_{WB}(s)\right) \quad (9)$$

*where $\Sigma'_{WB}(s) = 2s(\Sigma_{X_0} + \Sigma_{X_1} - Q_0 - Q_1) + Q_1 + Q_2 - 2\Sigma_{X_1}$ with*

$$Q_0 = \Sigma_{X_0}^{1/2}\left(\Sigma_{X_0}^{1/2}\Sigma_{X_1}\Sigma_{X_0}^{1/2}\right)\Sigma_{X_0}^{-1/2} \quad and \quad Q_1 = \Sigma_{X_0}^{-1/2}\left(\Sigma_{X_0}^{1/2}\Sigma_{X_1}\Sigma_{X_0}^{1/2}\right)\Sigma_{X_1}^{1/2}.$$

*Proof.* Combine Proposition 4 and Proposition 7.

## D   PERCEPTUAL SCALE AND FISHER INFORMATION

*Proof of Proposition 2.* One has $\mathbb{P}_{M|S}(m, s) = \mathbb{P}_{M|R}(m, \psi(s))$ which implies that

$$\frac{\partial \log(\mathbb{P}_{M|S})}{\partial s}(m, s) = \psi'(s)\frac{\partial \log(\mathbb{P}_{M|R})}{\partial s}(m, \psi(s)).$$

Therefore, by squaring this last equation and taking its expectation with respect to $M$ we obtain Equation (5). Then, taking the square root and integrating lead to the result. $\square$

# E EXPERIMENTAL DETAILS

For the GRF textures we used the following parameters (see distributions in Appendix A):

- Spatial freq. mode ($z_0$) : [0.5, 0.75, 1.0, 1.25, 1.5, 1.75, 2.0, 2.25, 2.5, 2.75, 3.0, 3.25, 3.5],

- Spatial freq. bandwidth ($b_Z$) : [0.25, 0.5, 0.75, 1.0, 1.25, 1.5, 1.75, 2.0, 2.25, 2.5, 2.75, 3.0, 3.25],

- Orientation bandwidth ($\sigma_\Theta$) : [1.5, 2.5, 3.5, 4.5, 5.5, 6.5, 7.5, 8.5, 9.5, 10.5, 11.5, 12.5, 13.5].

The code to generate these stimuli is available here[3].

For the naturalistic interpolation of textures we use the natural textures available here[4]. The interpolation parameters used are given in the main text (see Vacher et al. (2020) or Proposition 7 for the interpolation weight definition).

# F FISHER INFORMATION CALCULATION

**Pixel assumption** We consider that pixel are sample from a univariate Gaussian distribution. To compute the Fisher information, we estimate the mean and standard deviation and then use Proposition 4.

**Images assumption** We consider the image to be a stationary GRF. To compute the Fisher information, we estimate the power spectrum and then use Proposition 3.

**Wavelet assumption** We are using the steerable pyramid Portilla & Simoncelli (2000). Then, as in the case of VGG19 features, we consider that the vector of wavelet activations at each pixel location is a sample of multivariate Gaussian distribution. So we compute the empirical mean and covariance square root and use Proposition 4.

# G INDIVIDUAL PERCEPTUAL SCALES

In the figures below, we show the perceptual scale measured on each individual participant. Beware that the participants are not the same from one pair of textures to another despite being label as "Participant 0X".

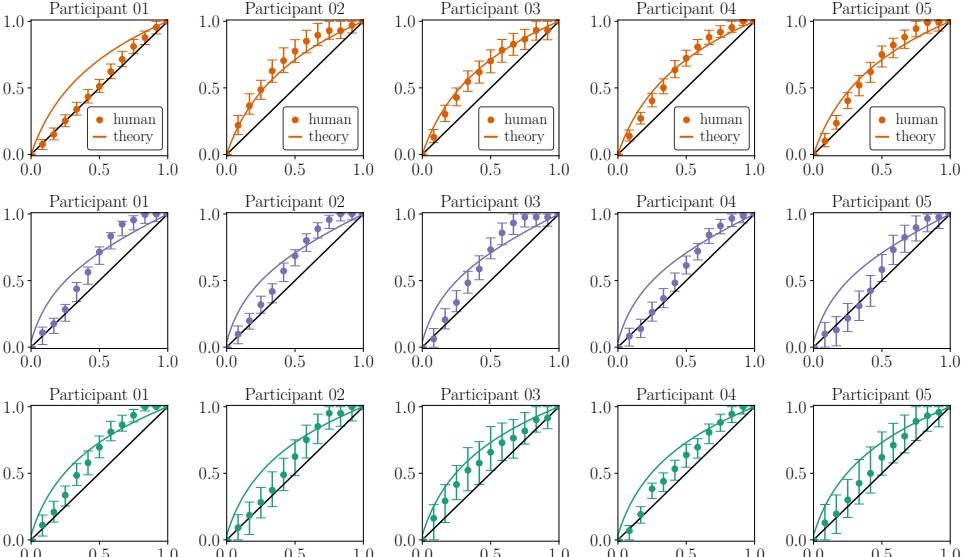

Figure 7: Gaussian textures. Measured perceptual scale for each participant. Top row: spatial frequency. Middle row: spatial frequency bandwidth. Bottom row: orientation bandwidth.

---

[3]https://

[4]https://

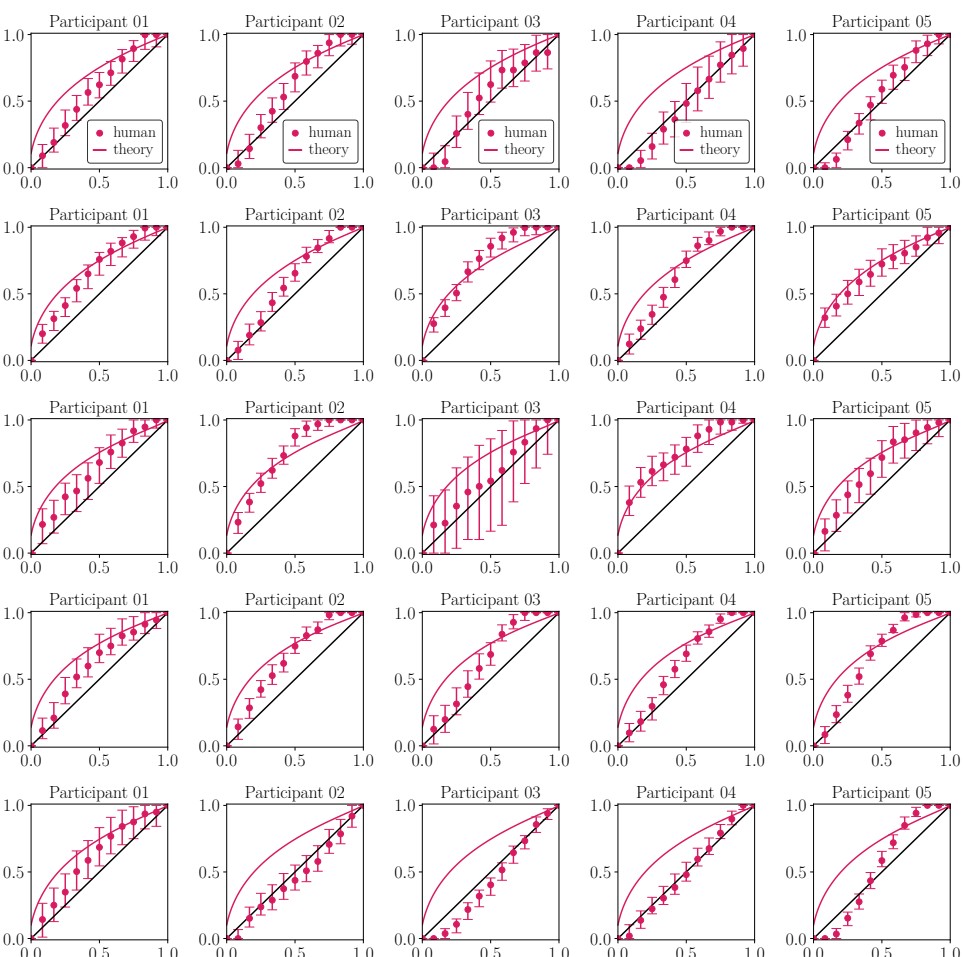

Figure 8: Naturalistic textures, early sensitivity. Measured perceptual scale for each participant. Top-to-bottom row: pair01 to pair05. Error bars represent 99.5% bootstrapped confidence intervals.

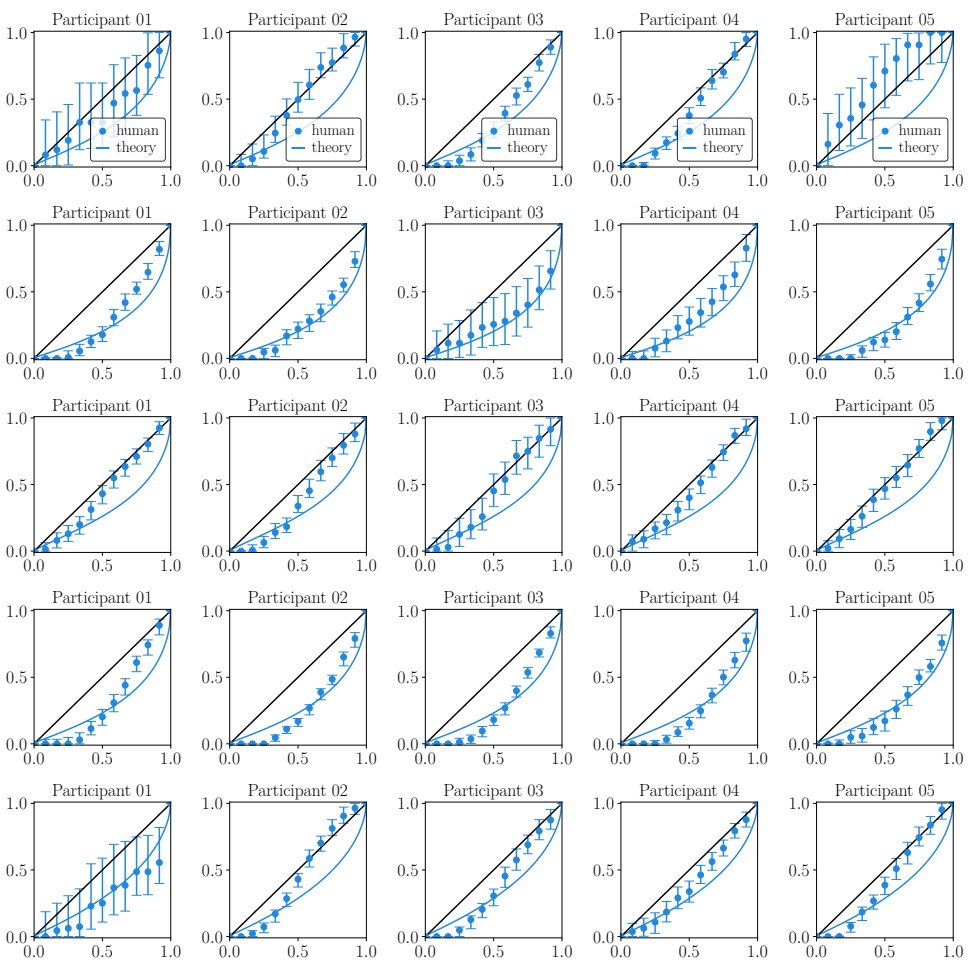

Figure 9: Naturalistic textures, late sensitivity. Measured perceptual scale for each participant. Top-to-bottom row: pair06 to pair10. Error bars represent 99.5% bootstrapped confidence intervals.

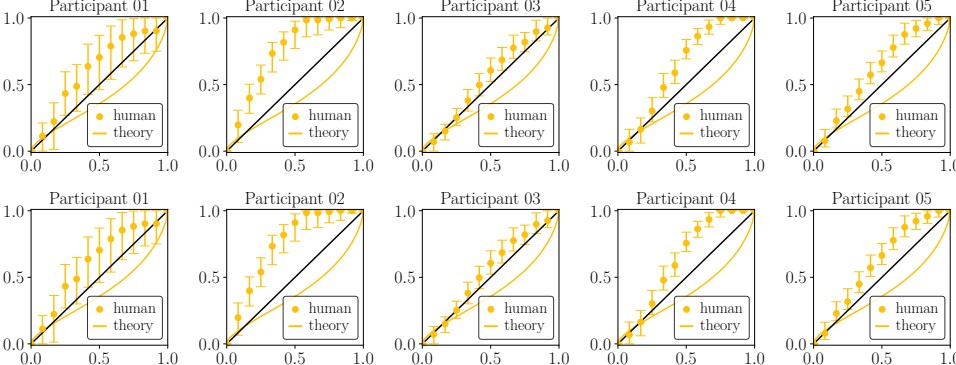

Figure 10: Naturalistic textures, conflicting prediction. Measured perceptual scale for each participant. Top and bottom row: pair11 and pair12. Error bars represent 99.5% bootstrapped confidence intervals.

## H ADDITIONAL EXPERIMENTS: POWER SPECTRUM NORMALIZATION

One conclusion of our first set of experiments on the set of texture pairs we have chosen is that the perceptual scale is mainly explained by the change in the power spectrum along the path from

one texture to the other. We decided to normalize those textures by the average power spectrum between both textures of a pair and to run again a set of experiment. This manipulation deteriorate the perceptual discriminability between both textures as one can observe in Figure 11. In addition, the manipulation changes the activation of VGG19 for each textures and for that reason we were not able to derive new predictions of the perceptual scale using the new VGG19 features. Instead we show in the result Figures 12 and 13 the initial prediction (whatever that means) also shown in Figure 2 of the main text. Surprisingly, some of the empirical results somewhat align with the predictions (Figure 12 `pair02` to `pair04` and `pair06` and `pair08` to `pair10`. We have no clue whether this observation is totally spurious or if it underlies that the perceptual scale is explained by the higher-order statistics of the textures captured by VGG19. Indeed, how would it be possible if we have not even computed the prediction corresponding to the presented image ? Well, VGG19 is trained on image-Net with some normalization, it is possible that the different layers of VGG19 are performing a normalization similar to the one we applied to the textures. Therefore, the predictions given by the original synthesized textures would be more correct that the one obtained with the power spectrum normalized textures. Unfortunately, the power spectrum manipulation also affects the level of noise estimated in the MLDS experiment (larger error bars in Figure 12/13 than in Figure 4/5) therefore the results are less reliable at the population level. Though, the individual results for the early and late sensitivity conditions shown in Figure 15, 14, tend to show that for some pair many participants have a similar perceptual scale (despite larger error bars) *e.g.* for pair01 : participant 03/04; for pair02: participant 01 to 03 and participant 05, etc. This is not the case with the conflicting prediction condition (yellow, Figure 13).

In conclusion, those preliminary results leave us with a lot of questions. A lot of work remain to be done to properly understand what features are able to explain the observed perceptual scale.

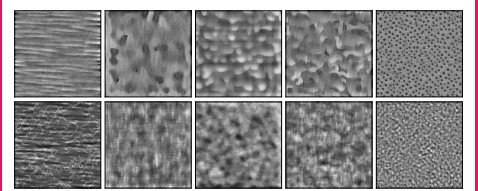 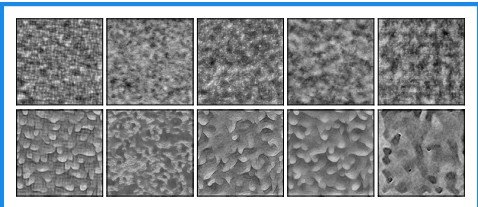

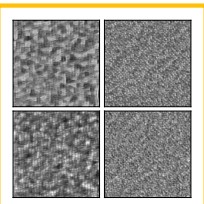

Figure 11: Texture samples normalized by their average power spectrum. Red corresponds to the early sensitivity group. Blue corresponds to the late sensitivity group. Yellow corresponds to conflicting prediction across VGG-19 layers.

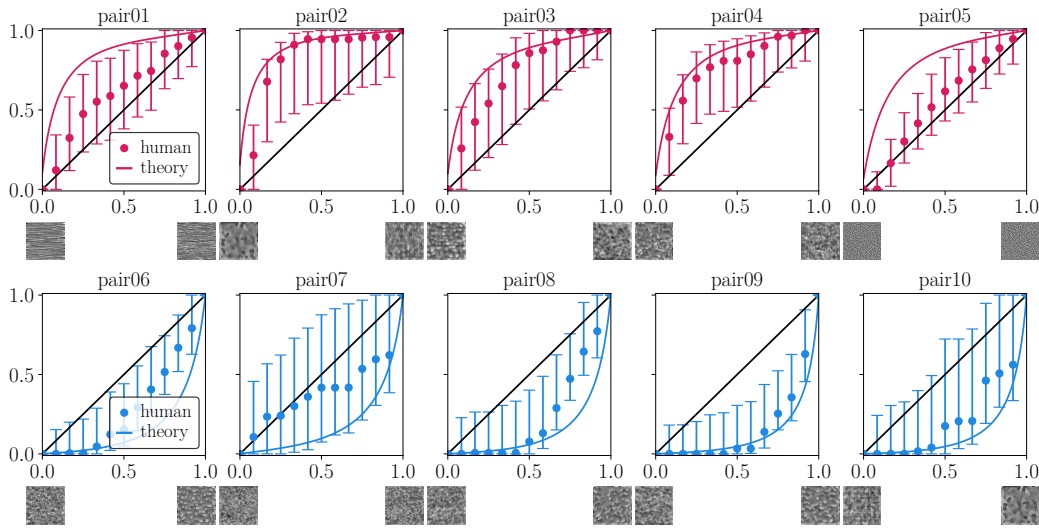

Figure 12: Measured and predicted (VGG19 averaged) perceptual scales for the early (top row) and late (bottom row) sensitivity pairs. Error bars represent 99.5% bootstrapped confidence intervals.

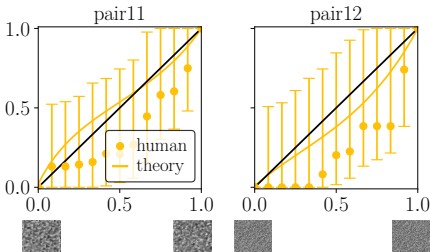

Figure 13: Measured and predicted (VGG19 averaged) perceptual scales for the conflicting prediction pairs. Error bars represent 99.5% bootstrapped confidence intervals.

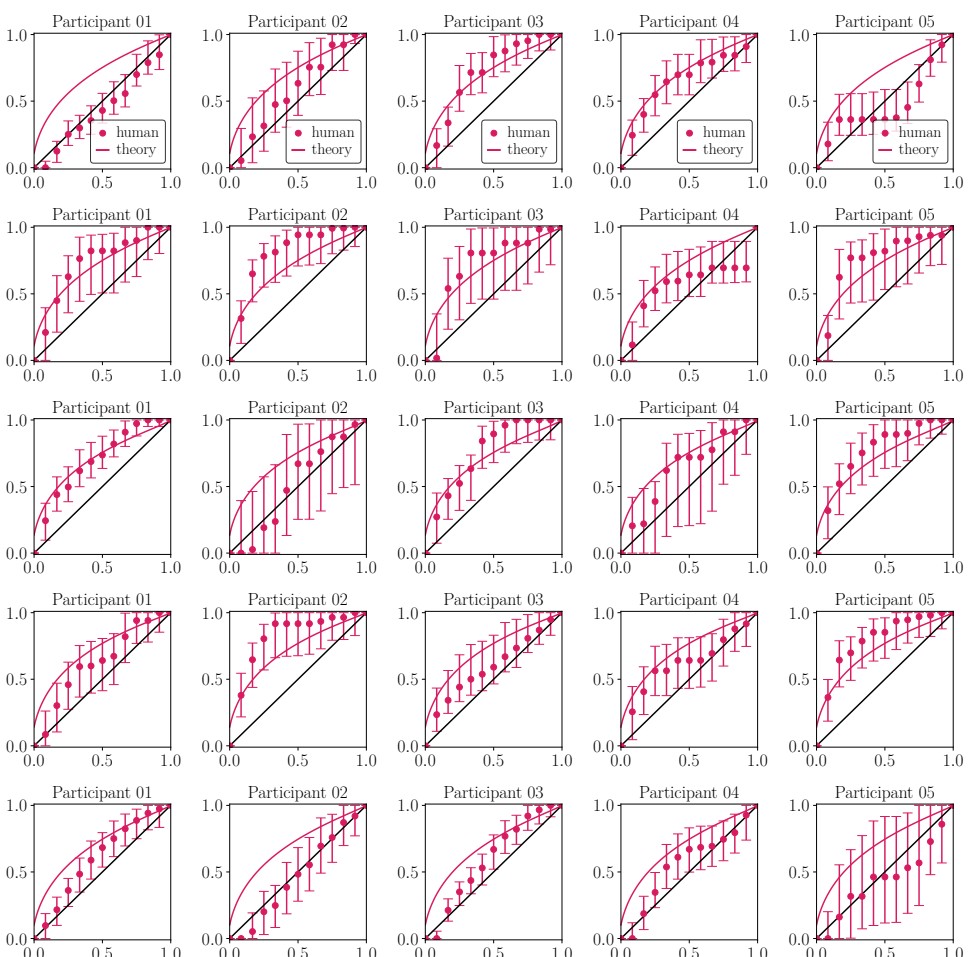

Figure 14: Naturalistic textures, early sensitivity. Measured perceptual scale for each participant. Top-to-bottom row: pair01 to pair05. Error bars represent 99.5% bootstrapped confidence intervals.

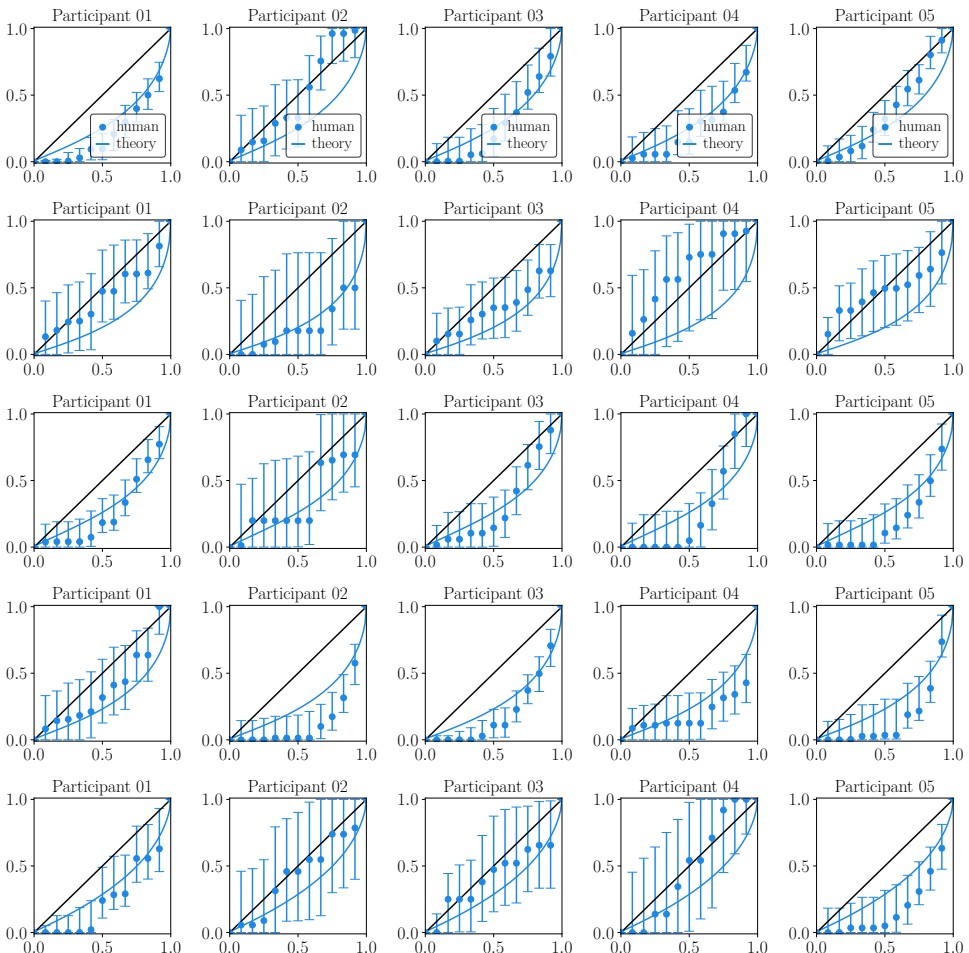

Figure 15: Naturalistic textures, late sensitivity. Measured perceptual scale for each participant. Top-to-bottom row: pair06 to pair10. Error bars represent 99.5% bootstrapped confidence intervals.

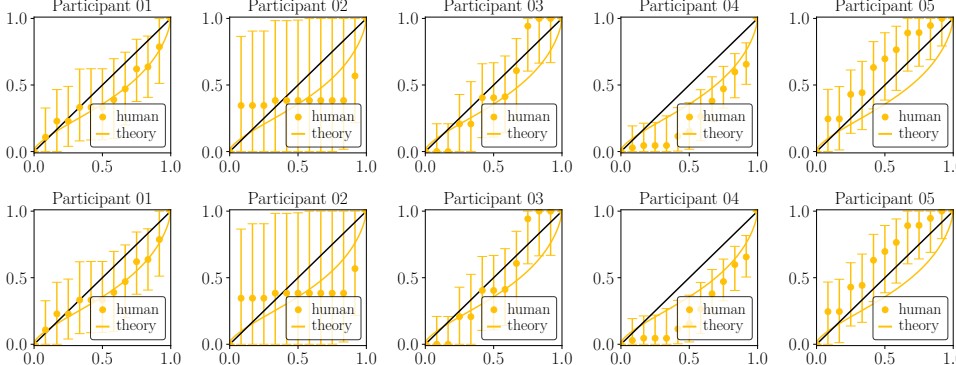

Figure 16: Naturalistic textures, conflicting prediction. Measured perceptual scale for each participant. Top and bottom row: pair11 and pair12. Error bars represent 99.5% bootstrapped confidence intervals.

