# OpenReview forum: "Perceptual Scales Predicted by Fisher Information Metrics"
_ICLR.cc/2024/Conference — ICLR 2024 poster_

### Official Review · Reviewer_qN8s · 2023-10-30

**Soundness:** 2 fair
**Presentation:** 2 fair
**Contribution:** 2 fair
**Rating:** 8
**Confidence:** 4

**Summary:**

The authors derive nonlinear relations between certain univariate image features and the perceptual scale (human response) to these features using Fisher information.
While the response for frequency and orientation of Gaussian Random Fields proves to be quite consistent with actual human responses derived from similarity judgements, the situation is more complicated in naturalistic textures.
For Gaussian fields the correspondence is good because (1) these images are easy to characterize with those parameters and hence, the Fisher information of those parameters captures all the information of the multidimensional objects, and (2) the discriminability is actually related to the image information. However, for the responses to interpolation between naturalistic textures, it is not clear which features to consider to compute the Fisher information. In this case, the authors try different features (pixel values, wavelet responses, VGG19 responses, and the power spectrum), and the best agreement is found using the power spectrum.

**Strengths:**

The authors address an interesting issue (the nonlinearity of the human image representation), and they show that Fisher information may be a driving factor of discriminability in simple textures defined by band-pass power spectra.
The provided expressions of Fisher information have technical interest.

**Weaknesses:**

It is not clear how to generalize their approach to more complicated textures, where it is not obvious the features to describe them and how to compute Fisher information, J, from those.

In the more complicated scenario the authors only try a limited (non conclusive) set of features on some so-called naturalistic textures which are not very general either. All this reduces the strength of the experiments and the possible conclusions. Moreover, in some of the cases (e.g. pixel and wavelet features) they do not explain how to compute J.

Given the normalization of the response axes, the derived nonlinear functions cannot be used as a metric to compute differences between textures.

**Questions:**

MAJOR QUESTIONS:

* Please give examples interpolating between more general textures (e.g. a brick wall, and a flower field, or a pile of fruits -see examples of natural textures in Portilla & Simoncelli IJCV 2000-). Do you get similar results (theoretical predictions and human responses) in those cases?

* The description of the method is confusing. For example, how do you get the nonlinear response in Fig. 1? I guess first you use the expressions in appendix A, then, you use \Psi in Eq. 6 and then you integrate, right?. This is not clear in the text. Similarly, how do you get the predictions in Fig. 2?. For the VGG response you assume they are Gaussian vectors, then you apply Proposition 4, and then you integrate Eq. 6?.

* Appendices give explicit expressions for the Fisher information for the frequency and orientation of Gaussian fields and of Gaussian vectors for the activations of VGG-19 (if they were Gaussian), but how do you compute J for the pixel representation? (just apply an FFT and then use that estimation of the spectrum and the formula in Preposition 3?).
How do you do it for wavelets?... What is the wavelet decomposition you used? There are many of them!

* The normalization of the response axes imply that all modifications of the stimuli lead to the same perceptual distortion. Simple visual inspection of the pairs in Fig. 2 shows that this is not correct. If the authors do not propose a way to give an absolute scaling in these different dimensions they should not claim that they are giving a metric of the image space.
What they just provide is a nonlinear (up to a scalar) relation between displacements in certain directions and variations in the inner representation, but this is not a metric.

* Following the above comment, the proposed method would be unable to give a measure to predict the Mean Opinion Score (MOS) on distortion in databases such as TID [Ponomarenko et al. 13] or KADID [Lin et al. 19]. If this is not the case, the authors should mention how to infer this MOS.

* The title is too vague, it does not reflect the content of the paper, and actually overstresses "distances" and "metrics" not quite addressed in the work. What about changing the title by something like: "Nonlinear image representations in humans from Fisher information".

MINOR ISSUES:

* In the abstract authors say "we demonstrate that it is related to the Fisher information of the generative model that underlies perception", while it should say "we demonstrate that it is related to the Fisher information of the generative model that underlies the stimuli"

* In the first paragraph of page 6 the authors say "We will see that in both cases..." ... Where do the authors show this? (this is related to the confusing description of the method stated above).

* Typos:
first paragraph of page 8 "VGG-19 to another (bottom-right of Fig. 2)" ---> "VGG-19 to another (bottom-left of Fig. 2)"
second paragraph page 8 "frequency mode (Proposition 2 and Proposition 2)" ---> ?

---

> ### Author Response · Authors · 2023-11-18
> **response 1**
>
> Thank you for reading our paper and for your detailled feedback. This very useful to us. We appreciate that you consider the question of non-linear representation of image interesting and that you are considering our results technically useful.
>
> **Response to weaknesses**
>
> > It is not clear how to generalize their approach to more complicated textures, where it is not obvious the features to describe them and how to compute Fisher information, J, from those.
>
> Parts of our results are generic in the sense our theoretical results hold as long as there is a generative model of your the image that is sufficient to compute the Fisher information (or estimate it numerically). We focus on the texture model that is capable of synthesizing a wide range of natural textures and that is often used in experimental vision studies (Gatys 2015, Wallis 2017, Vacher 2020, Ziemba 2021). The features we are testing are also the standard ones that are tested in vision studies.
>
> > In the more complicated scenario the authors only try a limited (non conclusive) set of features on some so-called naturalistic textures which are not very general either. All this reduces the strength of the experiments and the possible conclusions. Moreover, in some of the cases (e.g. pixel and wavelet features) they do not explain how to compute J.
>
> We acknowledge that we only run experiment with few textures. Though the goal of the current paper is to provide the technical results. We are already working in more general settings : generating image interpolations using deep generative models (eg VAEs) and evaluate the corresponding perceptual scale. So we expect to further extend the experimental work but this is beyond the scope of this paper. See the response to questions below for the calculation of Fisher information.
>
> > Given the normalization of the response axes, the derived nonlinear functions cannot be used as a metric to compute differences between textures.
>
> The reviewer is correct, the measured scale is relative and cannot be used to measure distances between textures. However, this is not our goal here. Yet, we explain that we can measure the discrepancy between model predictions and empirical perceptual scales using the AMS score introduced in section 3.2. Such a score is a measure of how much a model is aligned to perception.
>
> **Response to questions**
>
> > Please give examples interpolating between more general textures (e.g. a brick wall, and a flower field, or a pile of fruits -see examples of natural textures in Portilla & Simoncelli IJCV 2000-). Do you get similar results (theoretical predictions and human responses) in those cases?
>
>  In the supplementary material of Vacher *et al.* 2020, you can see that indeed a wide range of textures can be synthesized. We can definitely run a few more experiments before the camera ready deadline if our work were to be accepted (before the review period ends is not possible because of administrative delays).
>
>  > The description of the method is confusing. For example, how do you get the nonlinear response in Fig. 1? I guess first you use the expressions in appendix A, then, you use Ψ in Eq. 6 and then you integrate, right?. This is not clear in the text. Similarly, how do you get the predictions in Fig. 2?. For the VGG response you assume they are Gaussian vectors, then you apply Proposition 4, and then you integrate Eq. 6?.
>
> You are correct on everything. For Fig 1, it's Appendix A + Eq. 6 and for Fig 2, it's Proposition 4 + Eq. 6. We apologize if it's not clear we have clarified this in the Figures' caption and main text.
>
> > Appendices give explicit expressions for the Fisher information for the frequency and orientation of Gaussian fields and of Gaussian vectors for the activations of VGG-19 (if they were Gaussian), but how do you compute J for the pixel representation? (just apply an FFT and then use that estimation of the spectrum and the formula in Preposition 3?). How do you do it for wavelets?... What is the wavelet decomposition you used? There are many of them!
>
> We acknowledge that we did not give the details of the computation for all models. In every case, we use the Gaussian assumption so it's either Proposition 3 or 4. For the pixel measurement, we are looking at the pixels distribution (histogram). For the image assumption, it's in fact the GRFs assumption so here we compute modulus of the Fourier transform and Prop 3. For the wavelet, we use the steerable pyramid (Portilla & Simoncelli 2000).Then, it is similar to VGG, we consider that the vector of wavelet activations at each pixel location is a sample of multivariate Gaussian distribution so we compute the empirical mean and covariance.  We have added a section in the appendix with those details and point to this section in the Figure caption.

---

> > ### Author Response · Authors · 2023-11-18
> > **response  2**
> >
> > > The normalization of the response axes imply that all modifications of the stimuli lead to the same perceptual distortion. Simple visual inspection of the pairs in Fig. 2 shows that this is not correct. If the authors do not propose a way to give an absolute scaling in these different dimensions they should not claim that they are giving a metric of the image space. What they just provide is a nonlinear (up to a scalar) relation between displacements in certain directions and variations in the inner representation, but this is not a metric.
> >
> > Sorry for the confusion, we are not claiming that we are measuring a metric. In psychophysics, there is no absolute measure. Claiming to make an absolute measure of perception is highly controversial. That say, our point is that we are able to compare a theoretical "perceptual" scale as if a model was "perceiving" something to an empirical perceptual scale. We can compare both using the AMS score we introduce. So that we are comparing the displacement along a geodesic on the image manifold to how it is perceived. One could then claim to have matched the perception when the model prediction matches the empirical data.
> >
> > > Following the above comment, the proposed method would be unable to give a measure to predict the Mean Opinion Score (MOS) on distortion in databases such as TID [Ponomarenko et al. 13] or KADID [Lin et al. 19]. If this is not the case, the authors should mention how to infer this MOS.
> >
> > Thank you for pointing out KADID and TID databases. We did not know these databases. Using our framework we could compute a score from the square root of the Fisher information (around a specific distortion). This would again require some assumption about the measurements, so we could obtain scores with pixels, images, wavelet or vgg19 and so on. The idea is that the scores correlate with the image quality judgment in that if we are more sensitive to a perturbation the image will appear as more degraded (lower score). This is very interesting, though it is a bit different from our approach. The derivative of the perceptual scale is a measure of sensitivity at every single point. This is what we are trying to predict here.
> >
> > > The title is too vague, it does not reflect the content of the paper, and actually overstresses "distances" and "metrics" not quite addressed in the work. What about changing the title by something like: "Nonlinear image representations in humans from Fisher information".
> >
> > Thanks for the alternative title suggestion. The previous title we had was more like your suggestion so we decided to follow you suggestion of changing the title. After discussion among the authors, we deciced to use : "Fisher Information Predicts the Perceptual Scale of Texture Interpolation Paths".
> >
> >
> > **Minor**
> >
> > 1. Both are correct, it depends on your perspective. We would like to find the right generative model of the stimuli, the one that matches the perception of an observer ie the one that have better chances to be the one used by an observer.
> > 2. Right after Proposition 3.
> > 3. Corrected, thanks.
> >
> > **Extra references**
> >
> > Wallis, Thomas SA, et al. "A parametric texture model based on deep convolutional features closely matches texture appearance for humans." Journal of vision 17.12 (2017): 5-5.
> >
> > Ziemba, Corey M., and Eero P. Simoncelli. "Opposing effects of selectivity and invariance in peripheral vision." Nature communications 12.1 (2021): 4597.

---

> ### Comment · Reviewer_qN8s · 2023-11-23
> **I find the replies ok. I rise the score to 8 given the interest of the subject for ICLR (learning representations!) and quality of info-theoretic results**
>
> After rising my score to 8 (due to the replies, quality of the info-theoretic results, and interest of the subject for the ICLR community -learning representations!-), just a comment on related literature. Following this statement done in the repy to other reviewer: "To this date and up to our knowledge, we are not aware of any alternative (normative) models that provide predictions of the perceptual scales."... The authors may have missed other (normative) approaches based on manifold equalization (that includes goals such information maximization and error minimization) that lead to similar/equivalent nonlinearities.
> This concept dates back (in the univariate case) to:
> * Laughlin Phys. Biol. Proc. Im., Springer 1983 (for the luminance-brightness nonlinearity)
> * Twer & MacLeod Network 2001 (for nonlinear color responses)
>
> And has been extended to multivariate cases is:
> * Malo & Gutierrez Network 2006 (for nonlinear response to contrast of ICA-like patterns)
> * Laparra & Malo Front. Human Neurosci. 2015 (for color / contrast of PCA-like textures / energy of moving patterns)
>
> Check those and consider including references to these other normative/functional explanations of the nonlinearities. I think it would be nice to connect both methods in the future.

---

> > ### Author Response · Authors · 2023-11-23
> >
> > Thank you for the positive feedback !
> >
> > Laughlin 1983 : We think we have the same approach (or at least overlapping prediction). Previous works on information maximization has shown Eq. (1) of our manuscript ie that stimulus density is proportional the square root of Fisher Information. Though, we do not expand on this aspect.
> >
> > Twer & MacLeod Network 2001 : We were not aware of this work. After a brief read, the derivation is unclear to us. But, we will for sure consider it in the future. We are wondering if we can also derive results about Fisher information.
> >
> > Considering the two other references about higher-dimension, this is another line of work that we wish we will be able to tackle in the (less near) future. We will add these in the discussion. Predictions are more diverse when moving away from the univariate case but there is a lack of efficient experimental procedure to test it in human vision.

---

> > > ### Comment · Reviewer_qN8s · 2023-11-23
> > > **Quick comment on univariate vs multivariate methods**
> > >
> > > I see your point, but note that multivariate methods do not mean that you cannot compute non-linearities in unidimensional (univariate) directions. Actually, the comparisons with experiments are done in "unidimensional" examples.

---

> ### Author Response · Authors · 2023-11-23
>
> Right, we will need to go into the details !
>
> PS : do not forget to update your score in the main review (if you still wish to do so ofc).

---

### Official Review · Reviewer_wfqs · 2023-10-31

**Soundness:** 2 fair
**Presentation:** 2 fair
**Contribution:** 2 fair
**Rating:** 5
**Confidence:** 3

**Summary:**

The paper seems to measure the perceptual scale of spatial frequency, orientation, and synthetic texture interpolation.

**Strengths:**

1. The problem is of fundamental importance.

2. The reviewer likes reading the introduction, especially the literature review.

**Weaknesses:**

1. The reviewer has read the paper multiple times but fails to comprehend the main contributions.

2. The authors claim to have a convergence theorem, but what are the implications and practical relevance? Specifically, a Gaussian field is assumed, but high-dimensional natural images are highly non-Gaussian.

3. The function $\psi$ in Eq. (2) as the perceptual scale is tested in a very constrained scenario, without comparing to competing methods. For example, a recent computational model of the contrast sensitivity function considers spatio-temporal frequency, eccentricity, luminance, and area [C1].

4. Moreover, the experimental results regarding texture interpolation are performed in a discriminative setting, not from a generative perspective (i.e., texture synthesis).


[C1] stelaCSF: A unified model of contrast sensitivity as the function of Spatio-Temporal frequency, Eccentricity, Luminance and Area,
 SIGGRAPH 2022.

**Questions:**

1. The goals of the paper should be more precise.

2. The comparison to previous methods should be performed, and performed in a comprehensive way.

3. The authors may want to test their models on natural photographic texture images, besides the synthetic and simplistic ones (as shown in Figs. 1 and 2).

**Details Of Ethics Concerns:**

N.A.

---

> ### Author Response · Authors · 2023-11-18
> **response 1**
>
> Thank you for reading our manuscript and for your feedback. We appreciate that you liked the introduction and that you have a good opinion about the importance of our work.
>
> **Response to weaknesses**
>
> > The reviewer has read the paper multiple times but fails to comprehend the main contributions.
>
> Our work has three main contributions that are detailed in the last section of the introduction. We have updated the section to clarify our contributions. Here is a reformulation of these contributions :
>
> 1. Proposition 1 together with Proposition 3 allow to justify that the common assumption behind Bayesian modeling in psychophysics, that is the observer has a univariate representation of the distribution of the parameter of interest (*e.g.* spatial frequency), is compatible with the assumption that an observer is measuring spectral energy distribution of the image. To clarify, these assumptions correspond to a decision about the nature of measurement $M$ in the model given by Equation (2).
>
> 2. In the model given by Equation (2), the function $\psi$ is in fact measurable using the MLDS experiment. This has not been previously used in the context of this model so we are providing a clear link between theory and experiment. From the theory and with extra assumptions about what are the measurements $M$ we can derive predictions for the perceptual scale $\psi$. Thanks to contribution (i), in the case of GRF stimuli, assuming that $M$ are the whole image $F$ or the random variables $Z$ or $\Theta$ lead to the same prediction.
>
> 3. We propose to go further than just perceptual distances by comparing the model prediction of perceptual scale (which can be obtained as long as you have a generative model) to the empirical perceptual scales. This is deeper than distances because we are comparing the model behavior along a geodesic in the space of images to the perception of this geodesic. The AMS score introduced in Section 3.2 could serve as a measure of how well a model is aligned to visual perception.
>
> > The authors claim to have a convergence theorem, but what are the implications and practical relevance? Specifically, a Gaussian field is assumed, but high-dimensional natural images are highly non-Gaussian.
>
> The convergence theorem has been previously proven already. Here we explain how this result justifies the compatibility between different hypotheses made in Bayesian modeling of perception about an observer's measurement. See contribution (i) above.
> The result hold for stationary Gaussian images only. The question remains open for more complex images. Yet, we show that when this result does not hold it is necessary to make extra assumption about the observer's measurements and without the mathematical result, different assumptions could lead to different predictions  (Figure 6).
>
> > The function in Eq. (2) as the perceptual scale is tested in a very constrained scenario, without comparing to competing methods. For example, a recent computational model of the contrast sensitivity function considers spatio-temporal frequency, eccentricity, luminance, and area [C1].
>
> Thank you for pointing to stelaCSF model. We will cite it as an alternative line of work. The work is interesting, yet the goal is totally different from our. The stelaCSF model is a descriptive model (*i.e.* it does not provide an explanation for why the contrast functions are like they are) aiming at better detecting visual artifacts in computer vision applications such as Virtual and Augmented Reality. In contrast, our aim is to provide and test a normative model of perception (*i.e.* we assume that perception is driven by the inference of parameters from a generative model). To this date and up to our knowledge, we are not aware of any alternative (normative) models that provide predictions of the perceptual scales. Though, we are comparing different sub-assumption of our model about the nature of measurement $M$ (Figure 6b). In addition, the CSF is a measure of the limit of the perceptual system (between seen and not seen) while the perceptual scale is a measure of perceptual sensitive inside the window of visibility (Watson 1986)
>
>
> > Moreover, the experimental results regarding texture interpolation are performed in a discriminative setting, not from a generative perspective (i.e., texture synthesis).
>
> To be precise, we are actually synthesizing interpolated textures using a generative model (Vacher *et al.* 2020) and the generated textures are used in the psychophysical experiment.

---

> > ### Author Response · Authors · 2023-11-18
> > **response 2**
> >
> > **Response to questions**
> >
> > 1. Hopefully, we have clarify our contributions above and we will update the section accordingly.
> >
> > 2. We are not aware of any models capable of predicting the perceptual scale that we could compare our model to. We compare different sub-assumptions of our model about the nature of measurement $M$.
> >
> > 3. Ultimately the goal would be to test our model on natural photographic texture images. Yet, we have no generative model for true natural textures so we are using a state of the art model as an approximation. Note also that the images that you are judging as simplistic are not considered as such by the vision science community (e.g. Rust *et al.* 2005, Martinez-Garcia *et al.* 2019).
> >
> > **Extra References**
> >
> > Watson, A. B., Ahumada, A. J., \& Farrell, J. E. (1986). Window of visibility: a psychophysical theory of fidelity in time-sampled visual motion displays. JOSA A, 3(3), 300-307.
> >
> > Rust, N. C., \& Movshon, J. A. (2005). In praise of artifice. Nature neuroscience, 8(12), 1647-1650.
> >
> > Martinez-Garcia, M., Bertalmío, M., \& Malo, J. (2019). In praise of artifice reloaded: Caution with natural image databases in modeling vision. Frontiers in neuroscience, 13, 8.

---

### Official Review · Reviewer_AMLy · 2023-10-31

**Soundness:** 3 good
**Presentation:** 1 poor
**Contribution:** 2 fair
**Rating:** 5
**Confidence:** 2

**Summary:**

This paper shows that the assumption that an observer has an internal representation of univariate parameters such as spatial frequency or orientation while stimuli are high-dimensional does not lead to contradictory predictions when following a theoretical framework. The perceptual scale is found to correspond to the transduction function in this framework and is related to the Fisher information of the generative model underlying perception. The research suggests that the stimulus power spectrum largely influences the perceptual scale. Furthermore, the study proposes that measuring the perceptual scale can help estimate the perceptual geometry of images, going beyond simple distance measurements to understand the path between images.

**Strengths:**

- A theoretical analysis on the perceptual scale in the case of GRFs was performed.
- Different scaling experiments involving GRF and naturalistic textures were conducted.

**Weaknesses:**

- The theoretical analysis is performed for the case of GRFs, which does not apply to naturalistic textures (Note that Gaussian textures are a very limited class of images). Most of the Propositions are special cases of previous work, so what is the theoretical contribution of this work?
- The different scaling experiments only involve a small set of pairs and 5 naive participants. There may be insufficient data for detailed analysis, and actually results were presented in Sec. 3 without any analysis.
- The paper appears to be hastily written with many typos (e.g., page 5: e have -> we have; Fig. 2: the colors are wrong, and the caption's description conflicts with the main body, Page 8: Proposition 2 and Proposition 2; ...) and symbols are not always well-defined or explained.

**Questions:**

This looks like careful and sophisticated work at first glance. I did not notice major defects in the paper, to my knowledge. However, the paper is difficult to follow and would benefit from careful editing.
Since I do not have a solid background in this area, I cannot confidently evaluate the significance here.
It may be better if the authors can write their manuscript from the point of view of a general researcher in ICLR.

Additional question: Can the authors explain more on how measuring the perceptual scale helps estimate the perceptual geometry of images? This is claimed in the Abstract but rarely mentioned in the main body.

---

> ### Author Response · Authors · 2023-11-18
>
> We thanks you for reading our manuscript and for your feedback. We appreciate that you liked our theoretical approach.
>
> **Response to weaknesses**
>
> > The theoretical analysis is performed for the case of GRFs, which does not apply to naturalistic textures (Note that Gaussian textures are a very limited class of images).
>
> We acknowledge that from the point of view of computer vision, Gaussian textures allow to synthesized a very limited class of images, mostly what is known as micro-textures (see Galerne 2011). Yet, in vision study most experiments are performed using simple and/or deterministic stimuli such as drifting grating, moving dots, moving bars, sparse Gabor functions... Moving toward naturalistic stimuli or even better natural images is still out of reach as we are not yet able to make sense of how to control the generation of those images. We have tool to do so thanks to deep generative models but they work as a black boxes which is not satisfactory for vision experiment purposes.
>
> > Most of the Propositions are special cases of previous work, so what is the theoretical contribution of this work?
>
> We have made no new theoretical contributions to the fields of information theory (or more generally to mathematics) though we apply existing results to the theory of perception. In particular the consequence of Proposition 1 have never been explained previously. In the context of our paper, this shows that different hypothesis about the internal representation used by an observer leads to similar results though this is not always the case (*e.g.* see Vacher *et al.* 2018)
>
> > The different scaling experiments only involve a small set of pairs and 5 naive participants. There may be insufficient data for detailed analysis, and actually results were presented in Sec. 3 without any analysis.
>
> The number of participants is enough for this type of experiments (Devinck & Knoblauch 2012). We tested 15 pairs (3 in Fig.3, 10 in Fig. 4 and Fig. 5) which is a good sample size to start testing our model. The current work is not enough in term of experiment and we expect to test it further in a future work that will be more focused on experiments.
>
> > The paper appears to be hastily written with many typos (e.g., page 5: e have -> we have; Fig. 2: the colors are wrong, and the caption's description conflicts with the main body, Page 8: Proposition 2 and Proposition 2; ...) and symbols are not always well-defined or explained.
>
> We apologize for the typos and the wrong color in the captions. The second proposition 2 was refering to the one you can find in the appendix. This has been corrected, thank you ! Reviewer gWW2 has also highlighted that $s$ wasn't defined in Equation (1). This has also been corrected.
>
> **Response to questions**
>
> 1. ICLR is a wide and diverse community in which there is a subset that is interested in perception and is familiar with the cited literature. The goal of mentioning perceptual distances and geometry is to attract other people from the community who are interested in that topic but we would like to show the existence of tools to actually measure those distances (but we are doing more than that here) and that it is not sufficient to claim that a distance is a perceptual one. Too many papers are doing so without perceptual experiments.
>
> 2. Using texture interpolations (Vacher *et al.* 2020) we are controlling the change of the stimulus along a geodedic between two textures. Therefore, we are exploring a line on the manifold of natural textures (characterized by VGG feature statistics). The measure of perceptual scale reflects the sensitivity to this change. If we were to find a metric to interpolate between textures that leads to a linear perceptual scale, this would mean that we have found a representation of textures and a metric that is accounting for human perception (not the distance between textures but the sensitivity along the path joining those textures).
>
> **Extra references**
>
> Devinck, F., \& Knoblauch, K. (2012). A common signal detection model accounts for both perception and discrimination of the watercolor effect. Journal of Vision, 12(3), 19-19

---

> > ### Comment · Reviewer_AMLy · 2023-11-22
> >
> > Thank you for your response.

---

> > > ### Author Response · Authors · 2023-11-22
> > >
> > > Thank you for reading our response. Have it improve your opinion about our manuscript ?
> > > If not, could you elaborate on how we could write our manuscript "from the point of view of a general researcher in ICLR". Thank you.

---

### Official Review · Reviewer_gWW2 · 2023-11-02

**Soundness:** 2 fair
**Presentation:** 2 fair
**Contribution:** 2 fair
**Rating:** 3
**Confidence:** 3

**Summary:**

The paper concerns itself with perceptual measures, in particular the perceptual distance between images showing what to me look like "noise" images each with a dominant spatial frequency. A theoiry is developed, and then tested experimentally.

**Strengths:**

I very much approve of the style of the experiment.
And I applaud all efforts to measure human-based distances.

**Weaknesses:**

I am not fully convinced by the model - the departure from human measures is significant.

The stimulli are very limited - gray scale images of textures that look to me like band-limited noise.  I am not at all sure how I should generlise any reults in this paper to general images. That is, it is not clear the model is general.

I suppose what I would really like to see would be something akin to a just-noticable difference, and then to build a measure in frequency space based on jnds - by analogy to jnd in colour space.

--

Not all equations are not numbered, which is a presentation error because it makes discussion hard.

**Questions:**

What does the parameter "s" mean?

I do not understand the experiment. Three stimulli were presented with s1 < s2 < s3.  I suppose this means the parameter s somehow orders the stimulli, but I don't know how.  Then participants are required to pair either (s1,s2) or (s2,s3). Why remove the (s1,s3) option? (It's removal may bias results.)

Overall, I get the impression the the authors have conducted perceptually experiments (on a very low number of participants) and that this paper is probably better suited to one of the perceptual psychology forums.  But, I found it hard to read, so I could very easily be wrong.

---

> ### Author Response · Authors · 2023-11-18
> **response**
>
> We would like to thank you for reading our manuscript and for your feedback. We appreciate that you liked the efforts we put in collecting human perceptual data.
>
> **Response to weaknesses**
>
> > I am not fully convinced by the model - the departure from human measures is significant.
>
>  The core of the model is Equation (2) (see paragraph Fisher information in neural populations). The most recent development of this model has been done by Wei and Stocker (2017) who provide several correct predictions of previous psychometric measures. Our contribution is to use this model to predict the results of MLDS experiments which has been proposed back in 2003 by Maloney and Yang. The model qualitatively explains part of the observation (positive score in Fig. 6). Yet, there are indeed quantitative discrepancies between model and data (Fig. 4) that need to be addressed in future work. Note in addition, that our model has nothing to account for individual variability (see individual results that we added in the supplementary material)
>
>
> > The stimulli are very limited - gray scale images of textures that look to me like band-limited noise. I am not at all sure how I should generlise any reults in this paper to general images. That is, it is not clear the model is general.
>
>  We are using two types of texture stimuli.
>
> First, we are indeed using band limited noise stimuli (see Figure 1). Here, we know what to expect by changing the spatial frequency mode. Though, bandwidth parameters ($b_z$, $\sigma_\theta$) have never been tested before so even if the stimuli are somehow simple the experiments are new.
>
> Second, the textures used in the rest of our experiments are naturalistic textures synthesized by state-of-the-art texture synthesis algorithms (Vacher *et al.* 2020). This type of texture synthesis algorithms have been been widely used in vision studies (Freeman *et al* 2011, Wallis *et al* 2017, Vacher *et al* 2020, Ziemba *et al* 2021).
>
>
> > I suppose what I would really like to see would be something akin to a just-noticable difference, and then to build a measure in frequency space based on jnds - by analogy to jnd in colour space.
>
> Measuring a perceptual scale with MLDS is different from measuring JND (with other methods like staircase of 2AFC experiment). Yet, it provides similar but more detailed and precise information as the derivative of the perceptual scale is a measure of sensitivity (see Aguilar *et al.* 2017). One of our motivation for conducting this work is that we believe the MLDS method is powerful in providing complementary data that are important to depict the full relationship between images and human percepts.
>
>
> **Response to questions**
>
> 1. We apologize for not numbering all equations. We have updated the manuscript accordingly.
>
> 2. We apologize for introducing the parameter $s$ in Equation (1) without explaining what is is. The parameter $s$ is the stimuli. It was stated just before Equation (2). We have moved this description right after Equation (1).
>
> 3. We use the method of triads described in Knoblauch and Maloney (2008). The stimulus $s_2$ plays the role of a reference stimulus and the method does not introduce bias because all triplets (except non-informative ones) are tested in an experiment. We are not giving a lot of details about the experiment because we believe this is not the most interesting aspect of our paper as we are using a previously described protocol.
>
> 4. As few as 5 participants is considered appropriate for this type of psychophysical experiments (Devinck & Knoblauch 2012) . We provided the results for each individual in the supplementary material to illustrate inter-participant variability.
>
> **Extra References**
>
> Devinck, F., \& Knoblauch, K. (2012). A common signal detection model accounts for both perception and discrimination of the watercolor effect. Journal of Vision, 12(3), 19-19
>
> Wallis, Thomas SA, et al. "A parametric texture model based on deep convolutional features closely matches texture appearance for humans." Journal of vision 17.12 (2017): 5-5.
>
> Ziemba, Corey M., and Eero P. Simoncelli. "Opposing effects of selectivity and invariance in peripheral vision." Nature communications 12.1 (2021): 4597.

---

### Author Response · Authors · 2023-11-22
**Response to reviewer**

Hi Reviewers,

We have provided responses to all your comments. We would appreciate to hear back from you about whether or not it has improved your opinion about our manuscript. If not, we would appreciate to hear about what could improve our work ?

Thanks,

---

> ### Comment · Reviewer_gWW2 · 2023-11-22
> **Not moved me much**
>
> I thank you for your repsonses.
> For me though, it has not much moved my opinion.
> The model fit is not good, even to the very limited stimulii you have used.
> This raises considerable doubt over the ability to generalise.
> While I appreciate your comments regarding JND, I continue to argue that a perceptual measure in "texture space" should be premised on JND, because JND measures provide the equivalent of a unit ruler. Unit rulers are the basis of measurements.
>
> Personally I think perceprtual and subjective distances are under explored in general, but are very important.
> So although I think the work - or at least its description - falls short of publishable standard on this occasion,
> I wish you well and hope to read material from you in the near future.

---

> ### Author Response · Authors · 2023-11-22
>
> The MLDS method provides a unit ruler.
>
> The difference is $d_{i,j,k} = \vert{\psi(s_i)-\psi(s_j)\vert} - \vert\psi(s_j)-\psi(s_k)\vert $ and it is assumed to be corrupted by an internal noise $N_{\text{mlds}}$ of constant variance $\sigma_{\text{mlds}}^2$ *i.e.* $\Delta_{i,j,k} = d_{i,j,k} + N_{\text{mlds}}$. Therefore, the unit ruler in the MLDS framework is given by $\sigma_{\text{mlds}}$ (it is inferred from data but not shown) which can be related to observer's internal noise $\sigma$ or the standard dev measured in a 2AFC experiment *i.e.* JND. Though, we do not expand on that because this is not the key point of our work. Yet, it seems that we have skipped several steps of explanations to jump to what is of interest to us and that it prevents a naive reader from precisely understanding our work as noted by other reviewers.

---

### Meta-Review · Area_Chair_au99 · 2023-12-08

**Metareview:**

The paper proposes the theoretical derivation and empirical test of a psychophysical scaling method to measure perceptual space of low-level (visual) features, i.e. the mapping of the physical (image) space to a psychological representation internal to a human observer. The work extends previous theories derived for single stimulus parameter to a scaling method using somewhat naturalistic image textures. The projection here is that down the line this method (and likely improvements thereof) will allow us to establish perceptual spaces using scaling judgments based on realistic images.

The reviewers generally acknowledge the importance of the paper's topic. However, they are also generally concerned about the not so great fit of the model's prediction with the data, and the apparent limitations to use it for more natural textures (GRFs constitute a quite limited set of possible textures). Also, individual reviewers were missing a discussion and comparison to more traditional methods of measuring perceptual spaces (e.g. using JNDs) and previous literature concerned with scaling methods of textures. Finally, the clarity of the writing was generally deemed not great.

The AC's own impression is in large part matching the view of the reviewers. The topic is indeed important, in particular with regard to future comparisons between artificial and natural intelligent systems in terms of information representations and geometry. The proposed theory currently has limitations with regard to the assumed generative model of the textures, and its predictions are not well matched with empirical data. Nonetheless, it provides a first step in addressing the problem which likely will initiate further improvements of its approach. The writing could be clearer and in part more detailed where it mattered.

**Justification For Why Not Higher Score:**

The paper has too many shortcomings for granting a higher score, with the main concern being the limited support of the predictions from the empirical data.

**Justification For Why Not Lower Score:**

As mentioned above, it comes down to preference. The topic is interesting and relevant, and the paper provides a good first attempt for modeling and measuring perceptual spaces. Some of the current shortcomings of the predictions are due to limitations of the used generative models, which will likely be improved in future follow up work.

---

### Decision · Program_Chairs · 2024-01-16

Accept (poster)